# A brain-wide form of presynaptic active zone plasticity orchestrates resilience to brain aging in *Drosophila*

Sheng Huang[1,2☯], Chengji Piao[1,2☯], Christine B. Beuschel[1,2], Zhiying Zhao[1], Stephan J. Sigrist[1,2‡]*

**1** Institute for Biology/Genetics, Freie Universität Berlin, Berlin, Germany, **2** NeuroCure Cluster of Excellence, Charité Universitätsmedizin, Berlin, Germany

☯ These authors contributed equally to this work.
‡ Lead contact.
* stephan.sigrist@fu-berlin.de

**Data Availability Statement:** All relevant data are within the paper and its Supporting Information files.

## Abstract

The brain as a central regulator of stress integration determines what is threatening, stores memories, and regulates physiological adaptations across the aging trajectory. While sleep homeostasis seems to be linked to brain resilience, how age-associated changes intersect to adapt brain resilience to life history remains enigmatic. We here provide evidence that a brain-wide form of presynaptic active zone plasticity ("PreScale"), characterized by increases of active zone scaffold proteins and synaptic vesicle release factors, integrates resilience by coupling sleep, longevity, and memory during early aging of *Drosophila*. PreScale increased over the brain until mid-age, to then decreased again, and promoted the age-typical adaption of sleep patterns as well as extended longevity, while at the same time it reduced the ability of forming new memories. Genetic induction of PreScale also mimicked early aging-associated adaption of sleep patterns and the neuronal activity/excitability of sleep control neurons. Spermidine supplementation, previously shown to suppress early aging-associated PreScale, also attenuated the age-typical sleep pattern changes. Pharmacological induction of sleep for 2 days in mid-age flies also reset PreScale, restored memory formation, and rejuvenated sleep patterns. Our data suggest that early along the aging trajectory, PreScale acts as an acute, brain-wide form of presynaptic plasticity to steer trade-offs between longevity, sleep, and memory formation in a still plastic phase of early brain aging.

## Introduction

The aging of the human societies over the globe, with lifespan extension currently being unmatched by increases in healthspan, urges to better understand the molecular–cellular mechanistic underpinnings underlying the various age-associated alterations [1,2]. The current increase of human lifespan expectancy demands for (i) a better understanding of aging mechanisms [1–8] and (ii) safe paradigms to allow for healthspan extension [7,9–15]. While

**Funding:** This work was supported by grants from the Deutsche Forschungsgemeinschaft (DFG; German Research Foundation) to S.J.S. (SFB1315 TP A08, NeuroNex2, CoE NeuroCure and FOR2705 TP05). S.H. was supported by the Chinese Scholarship Council (201504910753) as well as by the Leibniz Association (SAW-2019-ISAS-4-SyMetAge). The funders had no role in study design, data collection and analysis, decision to publish, or preparation of the manuscript.

**Competing interests:** I have read the journal's policy and the authors of this manuscript have the following competing interests: S.J.S. has interests in TLL (The Longevity Labs), a company founded in 2016 that develops natural food extracts.

**Abbreviations:** AL, antennal lobe; ARM, anesthesia-resistant memory; ASM, anesthesia-sensitive memory; BRP, Bruchpilot; dFB, dorsal fan-shaped body; Dlg1, Discs large; MB, mushroom body; MTM, middle-term memory; Spd, spermidine; STM, short-term memory; Syn, Synapsin; Syx, Syntaxin; THIP, 4,5,6,7-Tetrahydroisoxazolo[5,4-c]pyridin-3-ol; wt, wild type; ZT, zeitgeber time.

the resilience against the effects of brain aging is an individual trait [16,17], its circuit, neuronal, and molecular basis remains insufficiently understood. Similarly, the resilience relevance of age-associated molecular and behavioral changes observed in humans and in animal models remains largely unclear [1–4,6,18], and discriminating protective from detrimental changes often remains speculative, impeding the development of safe healthspan extension paradigms.

Aging provokes changes in both sleep pattern and memory, which are suspected to functionally interact [5,6,19,20]. The fruit fly *Drosophila* has been widely used in discovering molecular, cellular, and circuit mechanisms of memory formation [21–23], as well as in understanding the mechanisms and functions of sleep [24–26]. Moreover, *Drosophila* has also been developed into a suitable animal model for studying aging and age-associated sleep pattern alterations [27–31] and memory decline [10,32]. With a relative short lifespan, but not too short, *Drosophila* allows to explore and causally connect age-associated molecular and behavioral changes that might pave the way for diagnosis and therapy.

Previously, a brain-wide form of presynaptic plasticity (PreScale) was described in *Drosophila*, which is triggered in the course of aging [8] and also acutely upon experimentally and genetically induced sleep loss [33]. PreScale is driven by up-regulations of the presynaptic ELKS family scaffold factor Bruchpilot (BRP), which in turn controls the levels of critical synaptic vesicle release factor (m)Unc13-family protein Unc13A at presynaptic active zones [8,34]. Genetically triggering PreScale by increasing *brp* gene copy number reduced memory formation [8] and evoked rebound-like additional sleep [33]. Speaking of healthy aging paradigms, supplementing with the body-endogenous polyamine spermidine (Spd) has prominent cardioprotective and neuroprotective effects across the aging of animal models [9]. Notably, Spd facilitates memory formation in mid-aged 30-day-old animals [10] and extends lifespan [15], and, at the same time attenuates the age-associated emergence of PreScale [8]. Similarly, bidirectional regulation of *mitochondrial Aconitase 1* was found to modulate PreScale-type plasticity in conjunction with its effects on lifespan and age-associated memory decline [35]. Thus, BRP-driven PreScale seems to operate at a critical intersection of different aspects of brain aging. Still, how exactly PreScale-type plasticity intersects with brain aging, importantly whether in a per se protective or detrimental manner, remained to be investigated.

We here provide evidence that PreScale operates at the intersection of principal organismal fitness and the formation of new memories during early aging. PreScale peaked at mid-age, changed sleep patterns during the early aging phase, and promoted lifespan but at the same time restricted the extent of forming new memories. Healthy aging paradigms (Spd supplementation, pharmacological induction of excessive sleep) apparently reset the need of triggering PreScale-type plasticity in the *Drosophila* brain. Thus, PreScale seemingly executes behavioral adaptations and trade-offs during a still plastic phase of early brain aging, illustrating how life strategy manifests on a circuit and synaptic plasticity level.

## Results

### Brain-wide presynaptic active zone plasticity (PreScale) peaks in the mid-aged *Drosophila* brain

In the nervous system, the operation of brain circuits relies on the proper fusion of neurotransmitter-filled synaptic vesicles at the so-called presynaptic active zone [34,36]. Our previous study described a new form of global presynaptic active zone plasticity upon early *Drosophila* aging [8], which here we refer to as PreScale. It is characterized by brain-wide increases in the levels of the ELKS-family core active zone scaffold protein BRP and BRP-associated release factors, particularly (m)Unc13-family protein Unc13A, the latter being crucial for synaptic vesicle release at *Drosophila* synapses [8,34,37–39]. BRP seems to operate as a master scaffold of the

*Drosophila* active zone, which recruits a spectrum of synaptic proteins to tune presynaptic active zone composition and consequently drives synaptic plasticity [33,40,41]. Intriguingly, a global increase of BRP levels associated with a corresponding increase in Unc13A was also observed upon sleep loss [33,41,42]. Here, the extent of sleep loss typically correlates with increases of BRP levels [42], while genetically enforced sleep reduces BRP levels below baseline levels [33].

We here first asked for the role of PreScale across the whole aging trajectory of *Drosophila*. So far, we had found that BRP/Unc13A levels were up-regulated at 30 days (30d) compared to 5 days (5d) of age [8]. Thus, we first quantified the levels of BRP and associated synaptic proteins across the *Drosophila* life history. We wondered what limits these presynaptic plastic changes might have and whether, for example, BRP would increase gradually across the whole fly lifespan.

Depending on the rearing condition, *Drosophila* can live up to about 3 months, which provides an opportunity to explicitly dissect age-associated molecular and behavioral changes (Fig 1A), with the chance to identify critical time windows of turning and switching events. We collected brain samples of wild type (*wt*) at the following ages: 1-day-old (1d), 10d, 20d, 30d, 40d, 50d, 60d, and 70d, for quantitative western blot analysis monitoring a spectrum of pre- and postsynaptic proteins: BRP [37,39], (m)Unc13 family protein Unc13A [34], synaptic vesicle protein Synapsin (Syn) [43], SNARE complex core component Syntaxin (Syx) [44], the homolog of mammalian scaffold protein PSD95 Discs large (Dlg1) [45], as well as the crucial autophagy protein Atg8a, which is regulated by the aging process [46]. Indeed, BRP, Unc13A, Syn, and Dlg1 first increased almost linearly to arrive at a plateau at 30d to 40d ("early aging") (Fig 1B–1F), while the ratios between activated, lipidated Atg8a (Atg8a-II) and inactive, unlipidated Atg8a (Atg8a-I) showed a clear trend of gradual reduction during early aging (Fig 1B and 1H), consistent with previous reports [10,35,46]. Thereafter, however, the levels of these proteins dropped gradually with further advanced aging (Fig 1A–1F). Syx levels also showed a principally similar behavior (Fig 1B and 1G). Independent experiments using another BRP antibody (Nc82) showed similar results (S1 Fig). Consistent with these quantitative western blot results (Fig 1A–1H), immunostaining identified a stronger increase relative to juvenile levels in BRP levels at early aging stages than at later advanced aging stages (Fig 1I and 1J). Nonlinear fitting via quadratic regression suggests maximum BRP, Unc13A, Dlg1, and Syn levels at an age between 30d and 40d in western blots (Fig 1K). Notably, the dropping of synaptic protein levels at advanced ages, most prominently BRP and Unc13A levels, appeared to be associated with declining survival rates (Fig 1A–1D and 1K). These results thus suggest a switch of PreScale at middle age and uncover biphasic synaptic changes across the *Drosophila* lifespan.

## PreScale dampens both spontaneous activity and membrane excitability of sleep-controlling dFB neurons

What is the circuit and brain activity relevance of early aging-associated presynaptic active zone plasticity we call PreScale? As a kind of master scaffold of the active zone, BRP drives the upscaling of associated synaptic proteins and controls the synaptic vesicle release properties, particularly release probability and release site number [33,34,37]. Genetically triggering PreScale by increasing *brp* gene copy number reduces memory formation and provokes rebound-like sleep patterns [33]. Furthermore, increasing from *wt* 2xBRP to 4xBRP at young age was shown to mimic early aging-associated PreScale-type plasticity and memory decline [8]. Thus, we chose to study the potential roles of age-associated PreScale in regulating brain activity by inducing PreScale genetically in young animals.

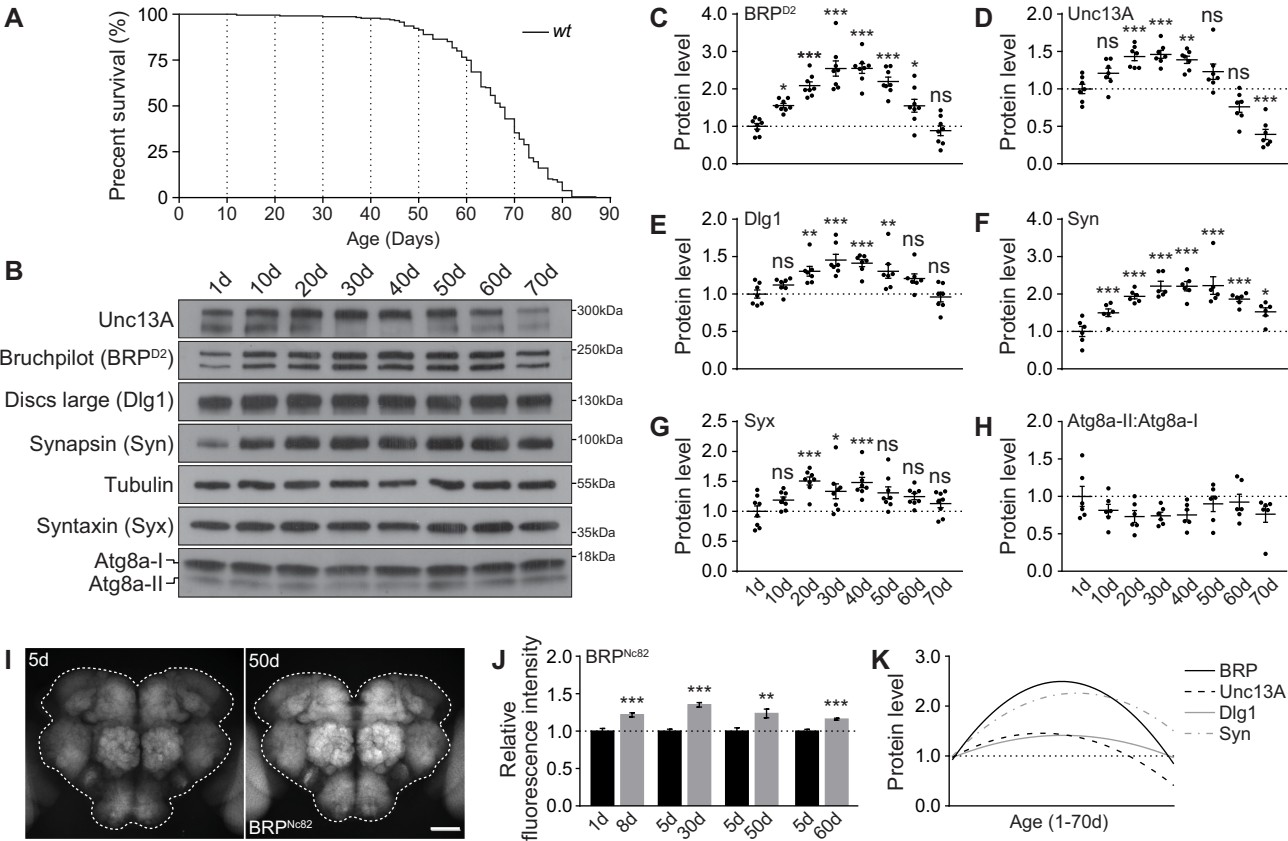

**Fig 1. Synaptic plasticity across the fly lifespan.** (**A**) A typical survival curve of *wt* female flies. *n* = 235. (**B-H**) Representative western blots (**B**) and relative levels of a spectrum of synaptic proteins, including BRP (**C**), Unc13A (**D**), Dlg1 (**E**), Syn (**F**), and Syx (**G**), and the relative ratios of active lipidated and unlipidated autophagy-related protein Atg8a (Atg8a-II:Atg8-I; **H**) in *wt* female flies across the lifespan. *n* = 6–8. One-way ANOVA with Bonferroni multiple comparisons test is shown. (**I** and **J**) Confocal images (**I**) and whole-mount brain staining analysis (**J**) of BRP in *wt* female flies at different ages. *n* = 15 for 8d, *n* = 9 for 30d, *n* = 14–15 for 50d, and *n* = 9 for 60d. Student *t* test is shown. *$p < 0.05$; **$p < 0.01$; ***$p < 0.001$; ns, not significant. Scale bar: 50 μm. Error bars: mean ± SEM. (**K**) Nonlinear line fitting quadratic regression of the protein levels of BRP (**C**), Unc13A (**D**), Dlg1 (**E**), and Syn (**F**) levels in western blot across the fly lifespan. Underlying data can be found in S1 Data. Raw images of this figure are provided in S1 Raw Images. BRP, Bruchpilot; Dlg1, Discs large; Syn, Synapsin; Syx, Syntaxin; *wt*, wild type.

To address this question, we first analyzed the activity status across the whole brain by utilizing the pan-neuronal driver *elav-Gal4* [47] to express the Ca²⁺-dependent activity reporter CaLexA [48], which is meant to be a proxy for the activity history of neurons. The expression pattern of the per se pan-neuronal *elav-Gal4* is known to be particularly strong in antennal lobe (AL) and mushroom body (MB) [47], and indeed *elav-Gal4*-driven CaLexA showed strong signals in AL and MB (Fig 2A). Interestingly, in 4xBRP animals, CaLexA signal within the ALs was significantly reduced while staying largely unaffected in MBs (Fig 2A and 2B). As the AL operates as the first synaptic relay station of the olfactory system and a central gate of olfactory sensory information processing [49], a generic drop in its activity levels might potentially support sleep.

PreScale might promote the transmission of specific activity patterns in relevant neurons in certain frequency domains, for example, the oscillatory activity pattern changes in the ellipsoid body R5 (previously called R2) network of sleep-deprived animals [50,51]. To analyze in a circuit-specific manner, we next examined the role of PreScale on the CaLexA signal of ellipsoid body R5 neurons using *R58H05-Gal4* [33,51]. We have shown recently that R5 CaLexA signal was up-regulated when *brp* gene copy number was increased from 2xBRP to 3xBRP,

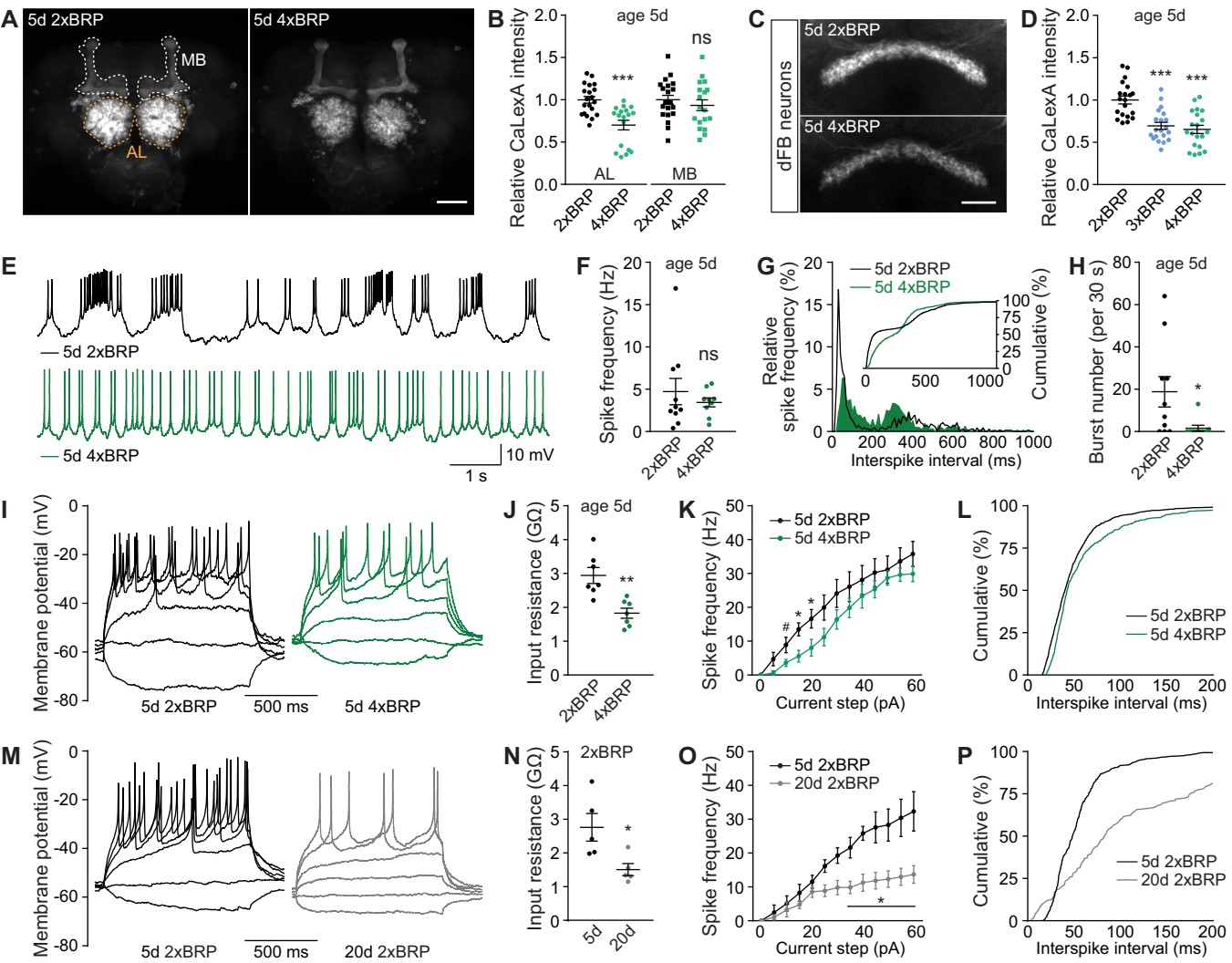

**Fig 2. PreScale plasticity provokes activity reprogramming in the dFB neurons.** (**A** and **B**) Confocal images (**A**) and whole-mount brain staining analysis (**B**) of CaLexA signal intensity with CaLexA expressed pan-neuronally by *elav-Gal4* in 2xBRP compared to 4xBRP female flies. *n* = 18–20 for all groups. Scale bar: 50 μm. (**C** and **D**) Confocal images (**C**) and whole-mount brain staining analysis (**D**) of CaLexA signal intensity with CaLexA expressed in dFB by *R23E10-Gal4* in 2xBRP compared to 3xBRP and 4xBRP female flies. *n* = 20 for all groups. Scale bar: 20 μm. (**E-H**) Spontaneous firing of dFB neurons in 5d 2xBRP and 4xBRP female flies, including representative membrane potential traces (**E**), mean firing rate (**F**), interspike interval distributions (**G**; Kolmogorov–Smirnov test, *p* < 0.0001), and bursting number for a 30-s window (**H**). *n* = 9–10 per group. (**I-L**) Voltage responses to 1-s current steps (from −10 to 60 pA, 5 pA) in dFB of 5d 2xBRP and 4xBRP female flies, including representative traces (**I**), input resistance (**J**), firing rates (**K**; two-way repeated-measures ANOVA analysis, *p* = 0.0245 for Current × Genotype interaction), and distribution of interspike interval at current steps till 40 pA (**L**; Kolmogorov–Smirnov test, *p* < 0.0001). *n* = 6–7 per group. (**M-P**) Voltage responses to 1-s current steps (from −10 to 60pA, 5pA) in dFB of 5d and 20d 2xBRP female flies, including representative traces (**M**), input resistance (**N**), firing rates (**O**; two-way repeated-measures ANOVA analysis, *p* < 0.0001 for Current × Age interaction), and distribution of interspike interval (**P**; Kolmogorov–Smirnov test, *p* < 0.0001). *n* = 5 for all groups. Student *t* test is shown for comparison between two groups. One-way ANOVA with Bonferroni multiple comparisons test is shown for comparisons of three groups. #*p* = 0.05; *p* < 0.05; **p* < 0.01; ***p* < 0.001; ns, not significant. Error bars: mean ± SEM. Underlying data can be found in S2 Data. AL, antennal lobe; dFB, dorsal fan-shaped body; MB, mushroom body.

representing a sleep-deprived state and higher sleep need [33]. Consistently, we here further show that 4xBRP animals also exhibited an increased CaLexA signal in R5 neurons (S2A–S2C Fig). Reducing *brp* gene copy number from 2xBRP to 1xBRP did not obviously change the CaLexA signal of R5 neurons (S2D and S2E Fig), likely in line with the only very mild reduction of sleep in 1xBRP flies [33].

The ellipsoid body R5 neurons were recently shown to receive excitatory inputs from helicon cells, within which a receptor of the inhibitory allatostatin-A peptide is expressed [52].

Furthermore, the dorsal fan-shaped body (dFB) neurons release allatostatin-A peptide to inhibit helicon cells, thus forming a negatively interconnected circuit with R5 neurons to regulate sleep (S2F Fig) [52]. Next, we expressed CaLexA in the well-studied sleep-promoting dFB using *R23E10-Gal4* [20,53]. Consistent with the inhibitory connection between dFB and R5 (S2B and S2C Fig) [33], we observed a strong decrease in CaLexA signal of dFB neurons in both 3xBRP and 4xBRP animals (Fig 2C and 2D). Unexpectedly, 1xBRP animals also showed significantly reduced CaLexA signal in dFB neurons (S2G and S2H Fig). dFB neurons have been shown to be electrically silent during wakefulness [53]. As CaLexA signal represents the activity history of given neurons [48], and 3xBRP and 4xBRP scenarios were shown to mimic a sleep-deprived state [33], we hypothesized that mechanically sleep-deprived animals might show reduced CaLexA signal in dFB neurons as well. To test this idea, we examined the dFB CaLexA signal of flies that have been sleep deprived for 12 h during nighttime. Indeed, a significant reduction in CaLexA signal in dFB neurons was observed in sleep-deprived animals (S2I and S2J Fig), similar to the 3xBRP/4xBRP scenario, suggesting reduced neuronal activity in dFB neurons during sleep deprivation. While the circuit activity states representing distinct forms of sleep behavior still need to be addressed in more detail, obviously, PreScale has the propensity of reprogramming the cumulative activity status of these sleep-regulating circuits.

As suggested by the CaLexA results in dFB at age 5d genetically triggered by PreScale (Fig 2C and 2D) and the well-characterized role of the dFB neurons in promoting sleep [20,53], we speculated that the plastic remodeling of neuronal activity and firing patterns in dFB neurons might be changed upon early aging. Moreover, if PreScale was really responsible to age-adapt dFB physiology, it might be possible to impose similar dFB changes in young animals through genetically installing PreScale. As 3xBRP and 4xBRP are essentially similar in dFB CaLexA (Fig 2C and 2D) and 4xBRP exhibited stronger sleep phenotype [33], we decided to perform in vivo electrophysiological recording of the dFB neurons in 4xBRP compared to 2xBRP at the age 5d (S2K and S2L Fig), measured their spontaneous spiking patterns, and injected current steps to elicit membrane voltage responses. Notably, though mean spontaneous spiking activity frequency remained unaffected, the regularity of spontaneous spiking was enhanced and bursting spikes were rarely seen in 4xBRP dFB neurons (Fig 2E–2H). Consistent with a recent study showing that the regularity of spontaneous firing of the sleep-regulating DN1p neurons determines sleep quality [54], the pattern changes of spontaneous firing of dFB neurons might contribute to sleep regulation in 4xBRP animals.

To determine the membrane excitability properties of dFB neurons comparing 4xBRP with 2xBRP, we injected current steps to depolarize the membrane and elicit action potentials. The dFB neurons in 4xBRP flies tended to be electrically silent, with lower input resistance, fewer action potentials elicited, and increased interspike intervals (Fig 2I–2L), in agreement with the CaLexA results (Fig 2C and 2D). To directly compare the effects of PreScale to potential effects of aging here, we intended to measure the electric properties of dFB at an age of 30d. Unfortunately, however, rates of successful patching were extremely low in these aged animals. However, at age 20d, we could successfully measure a few dFB neurons. The dFB neurons were indeed more electric silent, which showed reduced input resistance and increased interspike intervals, and current step elicited action potential firing almost failed to raise in 20d flies (Fig 2M–2P). While qualitatively similar, effect sizes were clearly more pronounced in the comparison of 20d to 5d than comparing 2xBRP to 4xBRP at age 5d. These data suggest that the genetically installed PreScale can operate as surrogate for important aspects of early aging and interventions of PreScale-type plasticity might be potential paradigms for healthy aging.

Taken together, our imaging and electrophysiology results point towards a reprograming of neuronal activity across the fly brain under PreScale-type plasticity, which likely reflects a shared synaptic origin of sleep homeostasis and early brain aging. The functional

reprograming of the sleep-promoting dFB/R5 network under PreScale suggests that PreScale might reprogram early aging-associated behavioral changes. We went on testing this hypothesis.

## PreScale promotes survival over memory formation during early aging

In our previous study, as also mentioned earlier, we could linearly increase BRP levels by titrating the number of *brp* genomic copies, which promotes sleep in a dosage-dependent manner (S3A–S3F Fig) [33]. Direct quantitative comparisons of BRP levels between early aging and genetic *brp* titration at age 5d by line-fitting indicate that per se both scenarios provoked BRP up-regulations to similar extents (Fig 3A). If establishing certain BRP and the associated changes in Unc13A levels had specific functions, especially throughout early aging, genetically titrating BRP levels might allow for uncovering mechanisms of age-associated behavioral changes.

To this end, we first monitored the longevity of flies under BRP-level titration and observed an extension of lifespan when increasing *brp* gene copy number stepwise from 1xBRP to 3xBRP. However, 4xBRP longevity suffered severely, being clearly shorter than in 2xBRP *wt* animals (Figs 3B–3E and S3L–S3V). Thus, BRP levels seemingly show an optimum curve here, with moderate increases in 3xBRP representing a level protecting the survival of animals from stress conditions, for example, in response to sleep deprivation [33,41,42,55] and during early aging (Fig 1).

If moderately elevated PreScale was indeed protective for survival, it might be particularly protective for flies challenged by chronic sleep loss and/or flies suffered from reduced lifespan. Thus, we established a 3xBRP situation in two sleep mutants, *wide awake* (*wake*) and *insomniac* (*inc*) mutants, characterized by reduced sleep [56–58]. Both mutants were recently shown to trigger PreScale likely in response to their sleep loss [33]. Indeed, 3xBRP rescued the lifespan of *wake* and *inc* male flies back to *wt* level (Fig 3F and 3H). At the same time, the survival of *atg7* autophagy mutants as well as of Alzheimer's disease model flies was not protected in both sexes (Fig 3J–3M). Interestingly, for female flies, 3xBRP only showed a tendency in lifespan extension in *wake* (Fig 3G), while the lifespan of *inc* was per se longer than *wt* and was not further extended by a 3xBRP background (Fig 3I). Because of the distinct sleep patterns between 5d female and male *wt* flies (S3G–S3K Fig), the sex difference in lifespan in *wake* and *inc* flies might be connected with the different impact from these mutations on sleep between male and female animals. Taken together, these data support a sex-specific, context-dependent role of PreScale in regulating lifespan.

As observed in model organisms and humans, memory formation and consolidation typically dwindle in their efficiency with advancing age [7,59,60]. Healthy aging, as an emerging concept [9,13], demands for not only longer life expectance but also specifically for a maintenance of life quality, importantly proper locomotor ability, as well as efficient sleep and memory formation abilities at older ages. Thus, we asked whether 3xBRP might be able to delay the normal age-associated memory decline, a phenomenon broadly reported for *Drosophila* [8,32,59]. Somewhat surprisingly, however, 3xBRP animals, while showing normal short-term memory (STM) at age 5d, displayed a significant reduction in STM at age 30d (Fig 3N and 3O). Middle-term aversive olfactory memory scores tested 3 h after training (3h MTM) were already significantly reduced in 3xBRP animals at age 5d (Fig 3P and 3Q).

Thus, moderately elevated PreScale in 3xBRP animals, while promoting lifespan in a context-dependent manner (Fig 3B–3M), at the same time decreased memory formation (Fig 3N–3Q). Given the dynamic nature of PreScale across the fly lifespan (Figs 1 and S1) and the lifespan extension of 3xBRP flies (Fig 3B–3E), an optimal level of PreScale during early aging

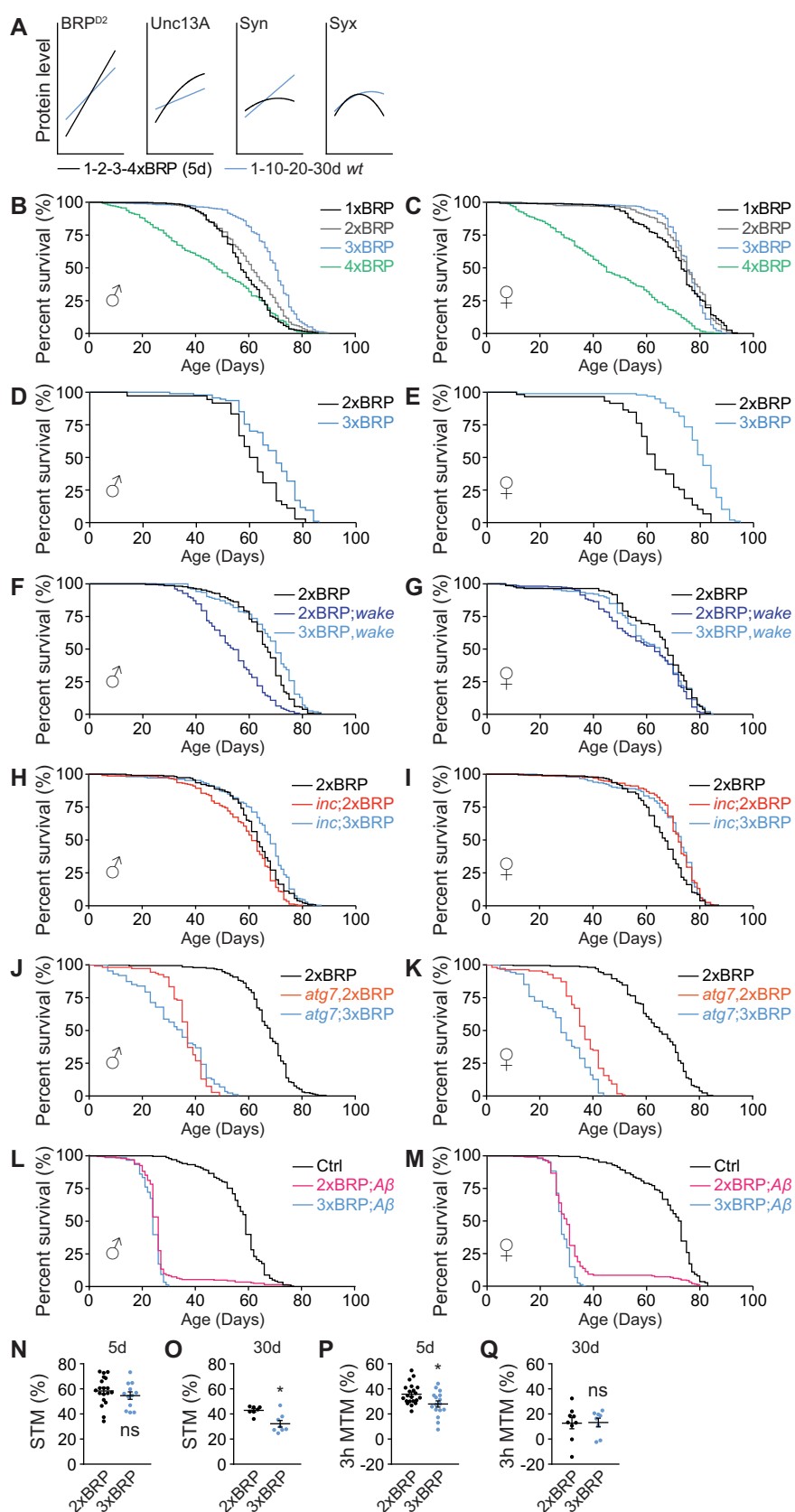

**Fig 3. PreScale promotes survival over memory formation during early aging.** (**A**) Direct curve-fitting comparisons between PreScale during early aging from 1d to 30d (Fig 1) and genetic *brp* titration from 1xBRP to 4xBRP at age 5d [33]. (**B and C**) Lifespan analysis of flies with BRP titration from 1xBRP to 4xBRP. For male flies (**B**), $n = 394$ for 1xBRP ($p < 0.001$), $n = 396$ for 3xBRP ($p < 0.001$) and $n = 395$ for 4xBRP ($p < 0.001$), compared to 2xBRP *wt* ($n = 393$). For female flies (**C**), $n = 400$ for 1xBRP ($p = 0.017$), $n = 399$ for 3xBRP (ns) and $n = 397$ for 4xBRP ($p < 0.001$), compared to 2xBRP *wt* ($n = 399$). (**D and E**) An independent experiment of the lifespan of 2xBRP and 3xBRP flies. For male flies (**D**), $n = 94$ for 3xBRP compared to 2xBRP *wt* ($n = 36$, $p < 0.001$). For female flies (**E**), $n = 98$ for 3xBRP compared to 2xBRP *wt* ($n = 59$, $p < 0.001$). (**F and G**) Lifespan analysis of flies with 3xBRP in *wake* mutant background. For male flies (**F**), $n = 234$ for 2xBRP;*wake* compared to 2xBRP *wt* control ($n = 233$, $p < 0.001$) or compared to 3xBRP,*wake* ($n = 155$, $p < 0.001$). For female flies (**G**), $n = 235$ for 2xBRP;*wake* compared to 2xBRP *wt* control ($n = 230$, $p < 0.001$) or compared to 3xBRP,*wake* ($n = 231$, ns). (**H and I**) Lifespan analysis of flies with 3xBRP in *inc* mutant background. For male flies (**H**), $n = 236$ for *inc*;2xBRP compared to 2xBRP *wt* ($n = 232$, $p = 0.009$) or compared to *inc*;3xBRP ($n = 212$, $p < 0.001$). For female flies (**I**), $n = 235$ for *inc*;2xBRP compared to 2xBRP *wt* ($n = 235$, $p < 0.001$, *wt* is also shown in Fig 1A) or compared to *inc*;3xBRP ($n = 182$, ns). (**J and K**) Lifespan analysis of flies with 3xBRP in *atg7* mutant background. For male flies (**J**), $n = 108$ for *atg7*,2xBRP compared to 2xBRP *wt* ($n = 195$, $p < 0.001$) or compared to *atg7*;3xBRP ($n = 87$, ns). For female flies (**K**), $n = 109$ for *atg7*,2xBRP compared to 2xBRP *wt* ($n = 208$, $p < 0.001$) or compared to *atg7*;3xBRP ($n = 87$, $p < 0.001$). (**L and M**) Lifespan analysis of flies with 3xBRP in Alzheimer's disease model flies (*elav>Aβ*). For male flies (**L**), $n = 203$ for 2xBRP;*Aβ* compared to *elav>mCD8-GFP* control ($n = 238$, $p < 0.001$) or compared to 3xBRP;*Aβ* ($n = 151$, $p = 0.006$). For female flies (**M**), $n = 249$ for 2xBRP;*Aβ* compared to *elav>mCD8-GFP* control ($n = 228$, $p < 0.001$) or compared to 3xBRP;*Aβ* ($n = 241$, $p = 0.002$). Gehan–Breslow–Wilcoxon test with Bonferroni correction for multiple comparisons is shown for all longevity experiments. (**N and O**) STM for 5d (**N**) and 30d (**O**) 2xBRP and 3xBRP flies. $n = 19$ for 5d 2xBRP, $n = 12$ for 5d 3xBRP. $n = 7$–8 for 30d for both groups at 30d. (**P and Q**) MTM tested 3 h after training for 5d (**P**) and 30d (**Q**) 2xBRP and 3xBRP flies. $n = 21$ for 5d 2xBRP, $n = 16$ for 5d 3xBRP. $n = 8$–9 for 30d for both groups at 30d. Student *t* test is shown. *$p < 0.05$; ns, not significant. Error bars: mean ± SEM. Underlying data can be found in S3 Data. BRP, Bruchpilot; MTM, middle-term memory; STM, short-term memory; *wt*, wild type.

might execute trade-offs to favor survival over complex and costly behaviors such as forming new memories [61,62].

## PreScale mediates early aging-associated alterations in sleep behavior

An age-associated decline of daily locomotor activities is typical for animal models and humans [63,64], which is often accompanied by sleep pattern changes [27,31,65]. Furthermore, PreScale seems to reprogram the activity of the interconnected R5/dFB network (Figs 2C–2P and S2A–S2J). If the age-associated BRP increase was causally connected to locomotor and sleep pattern changes, genetic attenuation of PreScale might result in a delayed onset of locomotor decline and sleep pattern changes during early aging.

We recently showed that removing an endogenous *brp* gene copy out of two by introducing a null mutation in the *brp* locus (referred to as 1xBRP) significantly reduces BRP levels and subsequently provokes a down-regulation of other crucial synaptic proteins, importantly Unc13A as well as Syntaxin [33]. Interestingly, we also observed that while 1xBRP animals displayed only slightly reduced total sleep under baseline condition, they showed facilitated wakefulness under mechanical disturbance at age 5d [33]. Here, we examined the daily locomotor activity and sleep patterns across early aging stages from age 5d over 20d and 30d to an age of 40d in 1xBRP compared to 2xBRP.

While a robust gradual reduction of total daily locomotor activity was evident for both 2xBRP control and 1xBRP flies during early aging, 1xBRP exhibited higher daily locomotor activity compared to 2xBRP animals across all ages tested (Fig 4A and 4B). This difference was most prominent at age 20d during daytime, especially at the light/dark transition at ZT12 (zeitgeber time) across the tested ages (Fig 4A and 4B). Notably, 1xBRP protracted early aging-associated changes relative to 2xBRP flies in the pattern and amount of daily locomotor activity, evident in 20d 1xBRP resembling 5d 2xBRP, 30d 1xBRP being similar to 20d 2xBRP flies, and so on (Fig 4A and 4B). In short, it seems that during early aging, 1xBRP animals mimic younger 2xBRP control flies concerning total locomotor activity. A similar but even more

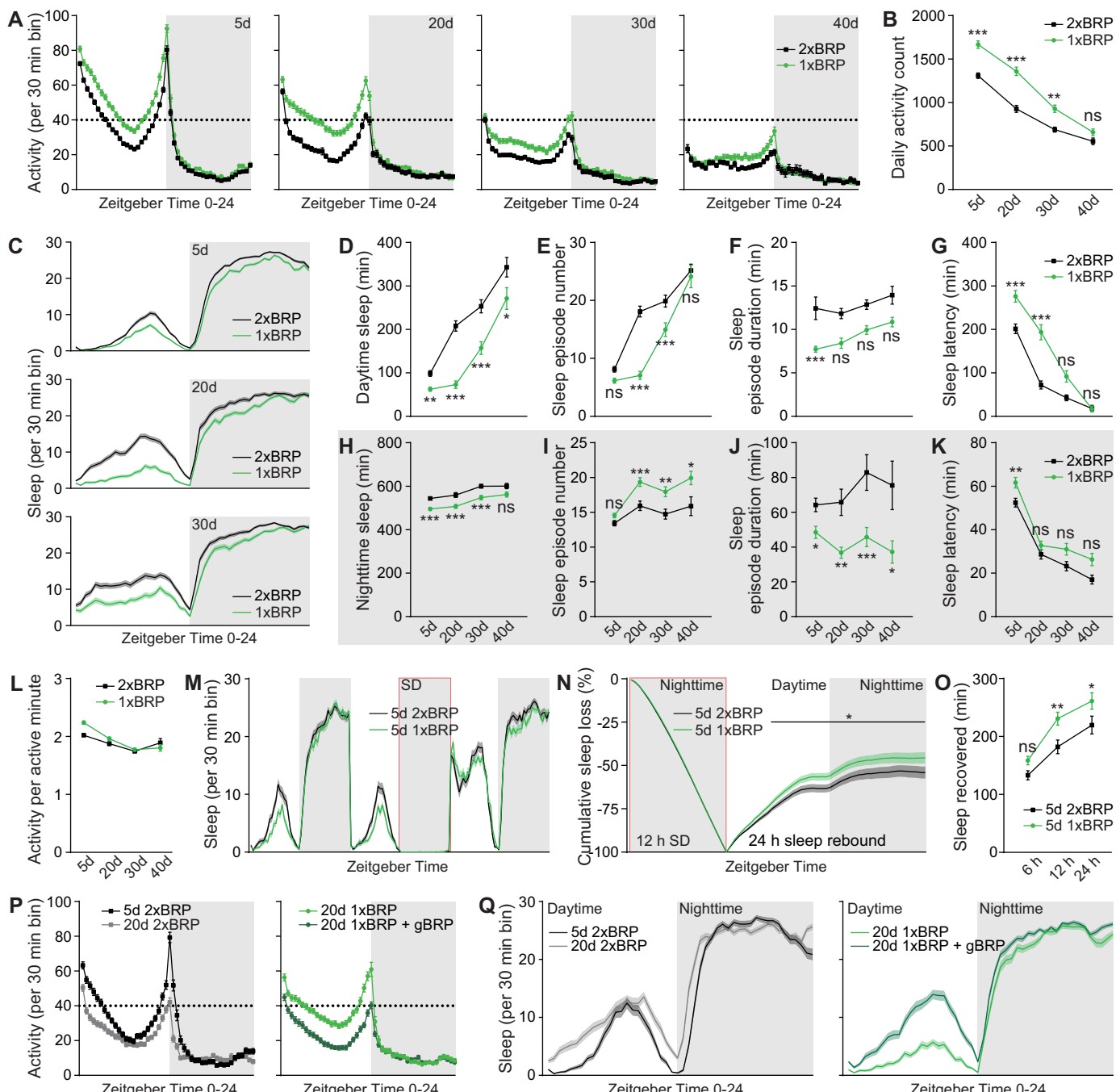

**Fig 4. PreScale mediates early aging-associated sleep pattern changes.** (**A** and **B**) Locomotor walking activity distribution across the day (**A**) and averaged daily total walking activity (**B**) of 1xBRP compared to 2xBRP female flies at ages 5d, 20d, 30d, and 40d. (**C-K**) Sleep structure of 1xBRP and 2xBRP female flies at ages 5d, 20d, 30d, and 40d averaged from measurements over 2–3 days, including sleep profile plotted in 30-min bins (**C**), daytime and nighttime sleep amount (**D** and **H**), number and duration of sleep episodes (**E, F, I,** and **J**), and sleep latencies (**G** and **K**). (**L**) Activity index (average activity count per active minute) of 1xBRP and 2xBRP female flies at ages 5d, 20d, 30d, and 40d. Correlation analysis did not identify any significant age-associated difference in activity index of both 1xBRP and 2xBRP animals. $n = 190–191$ for 5d, $n = 94–95$ for 20d, $n = 77–78$ for 30d, and $n = 32$ for 40d. Two-way ANOVA with Sidak multiple comparisons is shown. (**M**) Sleep profile for 5d 2xBRP *wt* control and 5d 1xBRP female flies for 3 consecutive days. (**N**) Normalized cumulative sleep loss during 12-h nighttime sleep deprivation and 24-h sleep rebound. $n = 116–121$ for both groups. Two-way repeated-measures ANOVA with Fisher LSD test detected a significant genotype × time interaction ($F_{(47, 11233)} = 3.933$; $p < 0.0001$) during sleep rebound. Asterisks above a line indicate time points at which the sleep percentage recovered differs significantly between female 2xBRP *wt* and 1xBRP flies, respectively. (**O**) Sleep recovered at three different time points after sleep deprivation for female 1xBRP compared to 2xBRP *wt* female flies at age 5d. $n = 114–119$ per group. Two-way ANOVA with Sidak multiple comparisons is shown. (**P** and **Q**) Locomotor walking activity (**P**) and sleep profile (**Q**) of 1xBRP female flies at age 20d rescued by introducing a transgenic *brp* copy (gBRP, *brp* P[acman]). $n = 63$–

64 for all groups. *$p < 0.05$; **$p < 0.01$; ***$p < 0.001$; ns, not significant. Error bars: mean ± SEM. Underlying data can be found in S4 Data. LSD, least significant difference; SD, sleep deprivation; *wt*, wild type.

pronounced effect was observed for the early aging-associated alterations of sleep pattern, especially daytime sleepiness, which was clearly suppressed in 1xBRP animals at all the ages tested (Fig 4C–4G). The suppression of daytime sleepiness was stronger at younger ages (i.e., 20d) evident in a substantial reduction of sleep episode number and duration (Fig 4C–4F). Of note, protraction of the decrease of sleep latency (at ZT0) was also observed for 1xBRP animals (Fig 4G). Nighttime sleep was also significantly suppressed by 1xBRP at different ages compared with age-matched 2xBRP control flies (Fig 4H). Interestingly, nighttime sleep was less consolidated and the difference in sleep latency at ZT12 diminished with age in 1xBRP animals compared to 2xBRP flies (Fig 4I–4K).

We speculated that if the 1xBRP constellation would attenuate the early aging-associated changes of sleep pattern, it might also show a stronger sleep rebound upon sleep deprivation, as aging is typically associated with a decline in sleep rebound in response to sleep deprivation [29]. To test this idea, we sleep-deprived both 1xBRP and 2xBRP flies at age 5d during nighttime for measuring sleep rebound afterwards (Fig 4M). In agreement with our speculation, 1xBRP flies indeed exhibited enhanced sleep rebound (Fig 4N and 4O). We here also performed an additional control experiment by putting an artificial genomic *brp* gene copy back to 1xBRP flies, which should restore *wt*-type sleep patterns. Indeed, the early aging-associated sleep pattern changes normally seen in 2xBRP *wt* flies were restored for these flies (Figs 4P, 4Q, and S4A–S4D), further supporting the specific role of BRP-driven PreScale-type plasticity in promoting early aging-associated sleep pattern changes and survival. At the first glance, an apparent paradox emerges in that 1xBRP animals concerning their sleep pattern regulation appear to age slower but are ultimately shorter lived, when compared to 2xBRP controls (Figs 3B, 3C and 4). Together with the extended lifespan of 3xBRP animals (Fig 3B–3E), this might be explained by the early aging-associated alterations of locomotor activity and sleep behavior being age adaptive and survival protective (Fig 4A–4L).

Contrary to 1xBRP (Fig 4), 3xBRP likely operates as a mimicry of the essential aspects of brain aging, which favors survival over cognitive functions (Figs 1, 2, 3, S1, and S3). Indeed, 3xBRP animals displayed a mild reduction in total locomotor activity (S4E and S4F Fig) and increased and consolidated sleep (S4G–S4O Fig) relative to 2xBRP *wt* controls at age 5d. The 2xBRP versus 3xBRP difference diminished gradually with age (S4E–S4O Fig). It appears that the genetically installed moderate PreScale increase in 3xBRP animals likely augments an early aging-associated brain reprogramming, which is important for later survival during advanced aging, as also suggested by the consequences of sleep loss early in life [66,67].

## Spermidine supplementation attenuates early aging-associated sleep pattern changes

Aging is typically associated with behavioral alterations affecting life quality and consequent burden of aging human societies [9,13]. Spermidine (Spd) supplementation has recently emerged as a promising paradigm to promote healthy aging, supported by studies ranging from flies and rodent models to human clinical trials [9]. In flies, Spd supplementation was shown to increase lifespan and attenuate early aging-associated memory decline [10,15,68]. Spd supplementation effects are based on metabolic alterations, age-protecting autophagic clearance, and mitochondrial respiration [10,15,68,69]. Importantly, we previously showed

that age-associated PreScale is efficiently suppressed by Spd supplementation during early aging stages (for example, at age 30d), while it does not affect PreScale in young animals [8]. These studies, together with our current findings (Figs 1 and S1), suggest that Spd supplementation might be able to delay the onset of age-associated behavioral alterations, importantly locomotion and sleep pattern changes (S5A Fig).

Indeed, the daily locomotor activity of Spd-supplied 30d flies was significantly higher when compared with age-matched control flies, while no difference was observed in 5d young animals, consistent with a solely aging-protective role of Spd supplementation (Fig 5A and 5B). Moreover, the early aging-associated sleep pattern changes were suppressed to a certain degree, especially concerning the daytime sleep phenotypes (Fig 5C–5G). Here, daytime sleep increases were substantially suppressed through a significant reduction in the number of sleep episodes with concomitant increases of sleep latency (Fig 5C–5E and 5G). Nighttime sleep changes were also suppressed mainly through an increase of sleep latency towards the values normally found in 5d young flies, while the number of nighttime sleep episodes and mean sleep episode duration stayed comparable to unsupplied siblings (Fig 5C–5G). All these changes were specific for 30d flies, as the sleep of 5d young flies was not affected by Spd supplementation (Fig 5A–5G), again consistent with the notion that Spd supplementation specifically affects the aging process [8,10,68,69]. Importantly, the typical age-associated reduction of sleep rebound upon sleep deprivation (Fig 5H–5J) was suppressed as well, allowing for a more efficient sleep compensation when reacting to sleep loss in 30d flies (Fig 5K–5M), but not in 5d flies (S5B–S5E Fig).

As mentioned earlier, we previously showed that Spd-supplied flies display suppressed PreScale during early aging [8]. Taken together, the effects of 1xBRP and Spd supplementation on early aging-associated sleep pattern changes and PreScale-type plasticity (Figs 4, 5, S4 and S5) [8] suggest that PreScale is at least partially responsible for early aging-associated sleep pattern changes. We suspect that, under Spd supplementation, the normal early aging-associated increase of PreScale and consequently behavioral changes might no longer be required, allowing for preserved memory and a delay of age-associated synaptic changes coupled with an alleviation of daytime sleepiness. Thus, it is tempting to speculate that metabolic reprogramming by Spd supplementation via rejuvenating autophagy and mitochondrial respiration [10,14,68,69] might suppress early aging-associated sleep patterns triggered by PreScale.

## Acute deep sleep reverses early aging-associated memory decline and PreScale-type plasticity

As mentioned earlier, in addition to sleep pattern changes (Figs 4, 5, S4, and S5), *Drosophila* aging is also associated with a cognitive decline evident in the reduced formation of aversive olfactory memories (Fig 6A and 6B) [10,32,59]. Thus, we intended to further address the intersection of longevity, the ability of forming new memories, and early aging-associated sleep pattern changes, in conjunction with the role of PreScale plasticity. If PreScale-encoded additional sleep during early aging served specific functions, and PreScale indeed represented an acutely operating active mechanism to promote such changes, then triggering excessive deep sleep in mid-aged flies might allow to reset these age-associated behavioral and molecular regulations. To test this, we turned to the acute feeding of sleep-promoting Gaboxadol (4,5,6,7-Tetrahydroisoxazolo[5,4-c]pyridin-3-ol, also called THIP), which has been shown to rescue memory formation in standard memory mutants and Alzheimer's disease model flies [70,71].

Before testing memory with different protocols of THIP treatment, we first confirmed the sleep-promoting effects of THIP by feeding 5d young *wt* flies with both a low concentration of

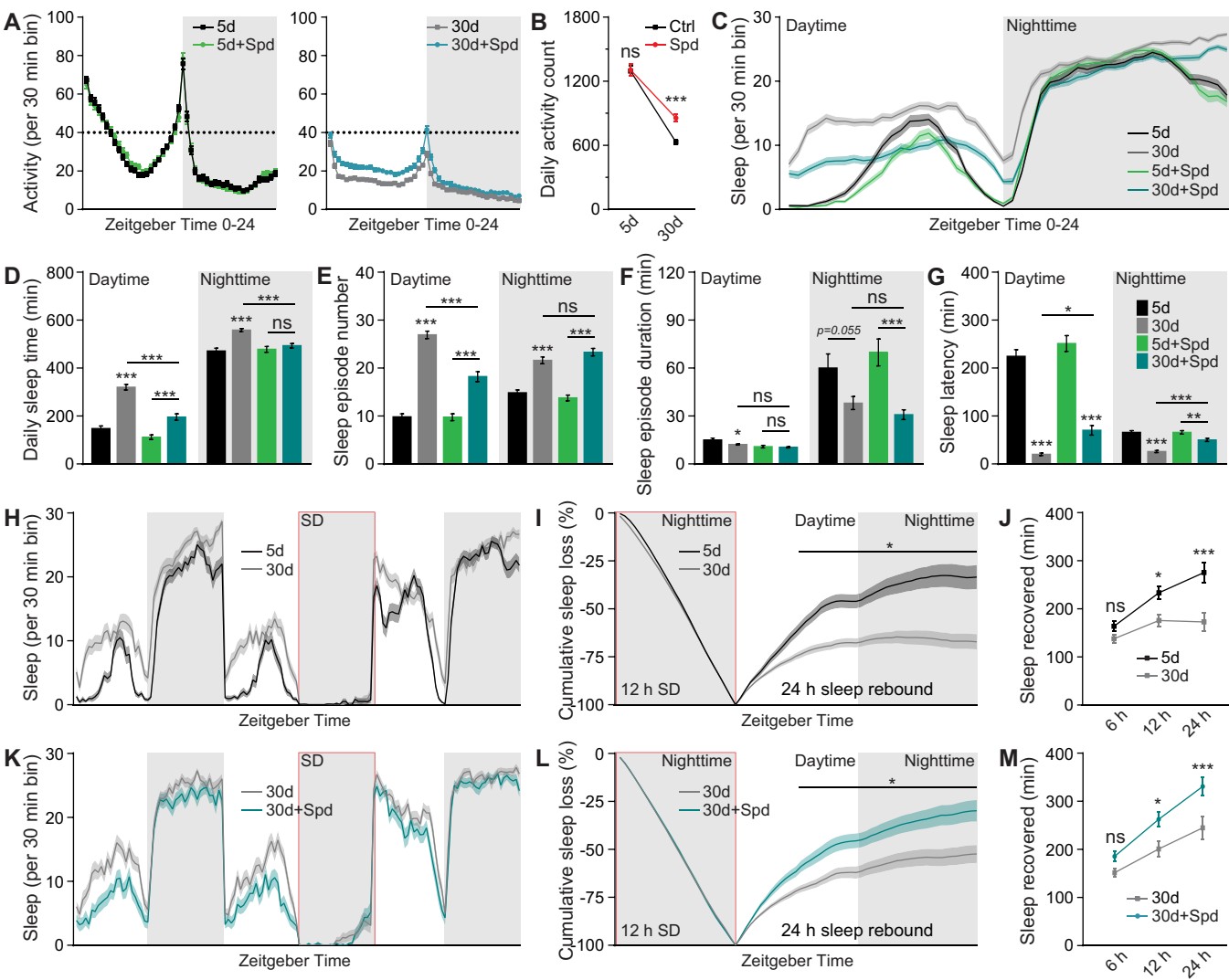

**Fig 5. Spd supplementation attenuates early aging-associated sleep pattern changes.** (**A**) Locomotor walking activity pattern of 5 mM Spd-treated *wt* female flies across the day at ages 5d and 30d. (**B**) Averaged daily locomotor walking activity of 5 mM Spd-treated *wt* female flies at ages 5d and 30d. Student *t* test is shown. (**C-G**) Sleep structure of 5 mM Spd-treated *wt* female flies at ages 5d and 30d averaged from measurements over 2–3 days, including sleep profile plotted in 30-min bins (**C**), daytime and nighttime sleep amount (**D**), number and duration of sleep episodes (**E** and **F**), and sleep latencies (**G**). $n$ = 93–96 for all groups. One-way ANOVA with Bonferroni multiple comparisons test is shown. (**H**) Sleep profile for *wt* female flies at age 30d compared to 5d for 3 consecutive days. (**I**) Normalized cumulative sleep loss during 12-h nighttime sleep deprivation and 24-h sleep rebound. Two-way repeated-measures ANOVA with Fisher LSD test detected a significant age × time interaction ($F_{(47, 5280)}$ = 4.227; $p < 0.0001$) during sleep rebound. Asterisks above a line indicate time points at which the sleep percentage recovered differs significantly between 5d and 30d *wt* female flies, respectively. (**J**) Sleep recovered at three different time points after sleep deprivation for 30d *wt* compared to 5d female flies. $n$ = 55–57 for both groups. Two-way ANOVA with Sidak multiple comparisons is shown. (**K**) Sleep profile for 30d *wt* female flies treated with 5 mM Spd compared to untreated for 3 consecutive days. (**L**) Normalized cumulative sleep loss during 12-h nighttime sleep deprivation and 24-h sleep rebound. Two-way repeated-measures ANOVA with Fisher LSD test detected a significant treatment × time interaction ($F_{(47, 4465)}$ = 7.168; $p < 0.0001$) during sleep rebound. Asterisks above a line indicate time points at which the sleep percentage recovered differs significantly between 30d 5 mM Spd-treated and untreated female flies, respectively. (**M**) Sleep recovered at three different time points after sleep deprivation for 30d 5 mM Spd-treated compared to untreated female flies. $n$ = 47–50 for both groups. Two-way ANOVA with Sidak multiple comparisons is shown. $^*p < 0.05$; $^{**}p < 0.01$; $^{***}p < 0.001$; ns, not significant. Error bars: mean ± SEM. Underlying data can be found in S5 Data. LSD, least significant difference; Spd, spermidine; *wt*, wild type.

0.025 mg ml⁻¹ (0.025THIP) and a higher concentration of 0.1 mg ml⁻¹ (0.1THIP) (S6A–S6F Fig) [70]. As feeding with 0.025THIP had already substantial sleep-promoting effects (S6A–S6F Fig), we first decided to feed young flies with 0.025THIP at age 2d for 2 days (48 h) and measure olfactory memory immediately thereafter (Fig 6C). After training flies using paired

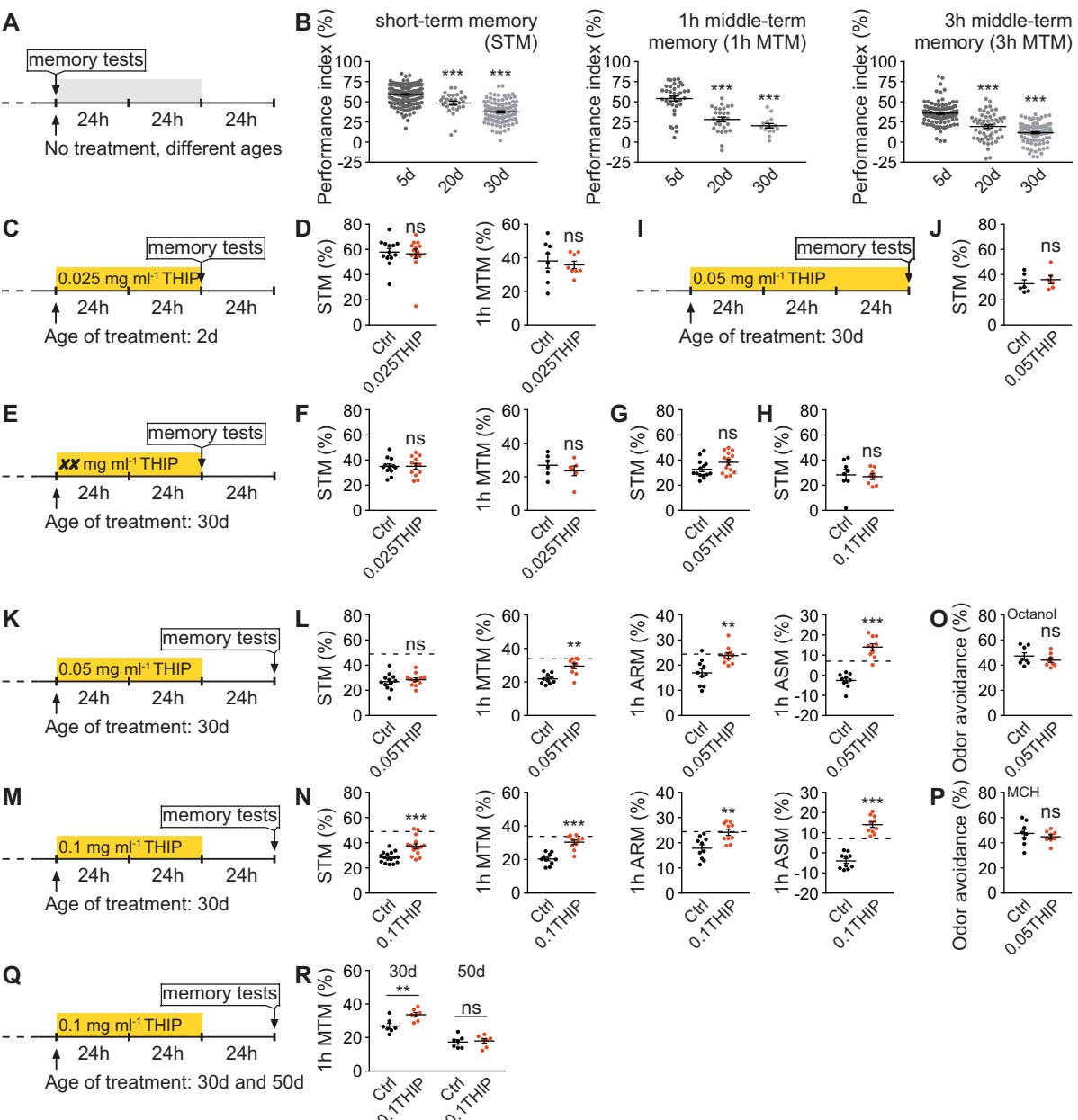

**Fig 6. Acute deep sleep reverses age-associated memory decline.** (**A** and **B**) Protocol for aging-related memory experiments (**A**) and typical age-associated memory decline in ages 20d and 30d *wt* flies (**B**), including STM, 1h MTM, and 3h MTM. (**C** and **D**) Protocol (**C**) for 0.025 mg ml⁻¹ THIP-treated *wt* flies at age 2d for 48 h and tested thereafter for STM and 1h MTM immediately, compared to untreated siblings (**D**). $n$ = 13 per group for STM, $n$ = 8 per group for 1h MTM. (**E** and **F**) Protocol (**E**) for 0.025 mg ml⁻¹ THIP-treated *wt* flies at age 30d for 48 h and tested thereafter for STM and 1h MTM immediately, compared to untreated siblings (**F**). $n$ = 12 per group for STM, $n$ = 6 per group for 1h MTM. (**G** and **H**) With the protocol in Fig 6E, two higher THIP concentrations 0.05 mg ml⁻¹ (**G**) and 0.1 mg ml⁻¹ (**H**) were used to treat *wt* flies at age 30d for 48 h and tested thereafter for STM immediately, compared to untreated siblings. $n$ = 14 per group for 0.05 mg ml⁻¹, $n$ = 8 per group for 0.1 mg ml⁻¹. (**I** and **J**) Protocol (**I**) for 0.05 mg ml⁻¹ THIP-treated *wt* flies at age 30d for 72 h and tested thereafter for STM immediately, compared to untreated siblings (**J**). $n$ = 6 per group. (**K** and **L**) Protocol (**K**) for 0.05 mg ml⁻¹ THIP-treated *wt* flies at age 30d for 48 h and tested 24 h after treatment for STM, 1h MTM, 1h ARM, and ASM, compared to untreated siblings (**L**). $n$ = 10–12 for all groups. (**M** and **N**) Protocol (**M**) for 0.1 mg ml⁻¹ THIP-treated *wt* flies at age 30d for 48 h and tested 24 h after treatment for STM, 1h MTM, 1h ARM, and 1h ASM, compared to untreated siblings (**N**). $n$ = 16 per group for STM, $n$ = 10 per group for 1h MTM, ARM, and ASM. (**O** and **P**) With the protocol in Fig 6K except that 2d *wt* flies were used, odor avoidance was tested for the two odors (3-Octanol (**O**) and MCH (**P**), i.e., 4-methylcyclohexanol) used for olfactory memory experiments. $n$ = 8 per group. Dashed lines indicate the average scores of respective memory components of 5d control *wt* flies without THIP treatment. (**Q** and **R**) Protocol (**Q**) for 0.1 mg ml⁻¹ THIP-treated *wt* flies at age 30d and 50d for 48 h and tested 24 h after treatment for 1h MTM compared to untreated siblings (**R**). $n$ = 7 per group. One-way

ANOVA with Bonferroni multiple comparisons test is shown. $^*p < 0.05$; $^{**}p < 0.01$; $^{***}p < 0.001$; ns, not significant. Error bars: mean ± SEM. Underlying data can be found in S6 Data. ARM, anesthesia-resistant memory; ASM, anesthesia-sensitive memory; MTM, middle-term memory; STM, short-term memory; THIP, 4,5,6,7-Tetrahydroisoxazolo[5,4-c]pyridin-3-ol; *wt*, wild type.

associative aversive olfactory conditioning, the animals were tested either right away for STM or 1 h later for 1h MTM. We did not find any differences in both STM and 1h MTM between 0.025THIP fed and unfed flies at age 2d (Fig 6C and 6D). It seems plausible that juvenile flies are still capable of forming memory at a maximum capacity and additional acute sleep might just not sufficient to boost olfactory memory formation. Thus, we continued by testing 30d flies, which displayed clearly reduced memories when compared to 5d young animals (Fig 6A and 6B).

However, feeding 30d *wt* flies with 0.025THIP for 48 h followed by immediate testing for olfactory memory did not show any improvement for both STM and 1h MTM (Fig 6E and 6F). We wondered if the concentration 0.025THIP was insufficient to provoke any beneficial effects and thus increased the concentrations to 0.05THIP and further to 0.1THIP. However, both concentrations did not provoke any significant improvement in STM (Figs 6E, 6G, 6H, and S6G–S6K). It should be said, however, that when feeding 0.05THIP with this protocol, STM scores tended to increase (Fig 6G), though not significantly. Another obvious protocol was to increase the duration of THIP feeding to 3 days (72 h), which, however, also did not improve STM (Fig 6I and 6J).

We next decided to feed 30d flies with 0.05THIP for 48 h but tested memories only 24 h *after* the end of THIP feeding (Fig 6K). Indeed, though STM was comparable to unfed control flies, 1h MTM was clearly improved over levels of unfed controls of the same age (Fig 6K and 6L). MTM is composed of two memory components that can be dissected through amnestic cooling, anesthesia-resistant memory (ARM) and anesthesia-sensitive memory (ASM) [22,23]. ARM is measured after amnestic cooling and ASM derives from subtracting ARM from MTM scores [22,23]. Notably, both ARM and ASM were fully restored by 0.05THIP treatment (Fig 6K and 6L). We further tested with a higher concentration of 0.1 mg ml$^{-1}$ THIP (0.1THIP) (Fig 6M) and found that STM was also improved, and 1h MTM, ARM, and ASM were all fully restored, just like under 0.05THIP treatment (Fig 6M and 6N). Importantly, THIP-feeding did not affect odor sensation, which is critical for olfactory memory formation (Fig 6O and 6P).

Next, we asked whether the same paradigm would suffice to reverse memory decline in flies during advanced aging (for example, 50-day-old), compared to flies during early aging (for example, 30-day-old). Importantly, 50d flies, which normally show a drastic decrease in 1h MTM, did not show any improvement upon 0.1THIP treatment when compared to 30d flies (Fig 6Q and 6R). Collectively, our results suggest that early aging is plastic and reversible and that the molecular and behavioral changes associated with this period are of adaptive nature in the realms of better survival. In contrast, brains of advanced age (50d) appear to lack plasticity and be rather nonreversible.

As shown above, a quasi-linear BRP increase (Fig 3A) is associated with the occurrence of early aging-associated dFB activity reprogramming, sleep pattern changes, memory decline, as well as longevity (Figs 1–6). Furthermore, Spd supplementation suppresses both sleep pattern changes (Figs 5 and S5) and PreScale during early aging [8] and promotes memory and longevity [8,14,15]. We wondered if acute THIP-feeding, which rescued memory decline (Fig 6K–6N), might also suppress the early aging-associated PreScale indicated by elevated BRP levels and sleep patter changes, similar to that of Spd supplementation (Fig 7A). Indeed, immunohistological examination revealed a clear reduction of BRP levels in 0.1 mg ml$^{-1}$ THIP-fed 30d flies when compared to unfed siblings (Fig 7B–7D). Notably, this was specific to age 30d flies,

as BRP levels were not different in 0.1THIP-fed 5d brains from nonsupplemented sibling controls (Fig 7B–7D). Sleep pattern changes were also suppressed by 0.1THIP treatment tested 1 day after the treatment specifically in 30d animals (Fig 7E–7L), but not in 5d young animals (S7 Fig).

If the suppression of PreScale (Fig 7A–7D) was causal for the restoration of memory in THIP-treated mid-aged 30d animals (Fig 6K–6R), increasing *brp* gene copy number from 2xBRP to 3xBRP might attenuate the improvement in memory formation. Thus, we again utilized the same THIP paradigm (Fig 7M and 7O) for comparing 2xBRP with 3xBRP animals at either age 2d or 30d. While 0.1THIP treatment did not show any obvious effect in 2d animals (Fig 7N), it consistently improved 1h MTM of 30d 2xBRP. In contrast, no such benefit was observed for 30d 3xBRP animals (Fig 7P). Thus, these data suggest a causal relationship between reducing PreScale and being able to effectively form new memories for mid-aged flies.

Motivated by the suppression of early aging-associated PreScale via THIP and Spd supplementation (Fig 7A–D) [8], and the longevity extension effect provoked by Spd supplementation [14,15], we tested the lifespan of *wt* flies fed with 0.05 mg ml$^{-1}$ THIP chronically starting at age 2d (Fig 7Q). While *wt* male flies showed a slight tendency towards lifespan extension (Fig 7R), female flies lived significantly longer upon 0.05THIP-feeding compared to unfed siblings (Fig 7S). The sex-specific difference here might be explained by the lower baseline sleep levels of females and the consequent greater dynamic range of the THIP-mediated sleep promotion (S3G–S3K Fig). In summary, the induction of acute, extensive deep sleep in mid-aged flies seemingly allows for a reset of processes underlying and driving brain aging, in effect also promoting lifespan, though in a sex-specific manner.

Taken together, we here used two distinct rejuvenation paradigms, Gaboxadol/THIP treatment and Spd supplementation, which provoked concordant effects: restoration of memory formation, increased longevity, and, importantly, making PreScale dispensable in the brains of mid-aged flies (Figs 5, 6 and 7). While we propose that PreScale is formed "on demand" in response to neuronal physiological state changes that build in response to normal aging and are susceptible to THIP and Spd, PreScale and subsequent trade-offs might be no longer "on demand" under rejuvenation paradigms. Given the roles of PreScale in controlling early aging-associated sleep pattern changes and longevity (Figs 1, 3 and 4), this form of plasticity emerges at the intersection of sleep, longevity, and age-associated memory decline. Notably, PreScale is also reversibly triggered by sleep deprivation [33,41,42], suggesting a generic and acute requirement of attenuating PreScale in resetting and denoising of circuits.

## Discussion

Resilience designates the capacity of a system, enterprise, or person to maintain the core purpose and integrity in the face of drastically changing circumstances [16,72]. Resilience in human physiology is the capacity of adaptation in counteracting stress and adversity, while maintaining normal psychological and physical functions [73]. A key result of stress probably lies in the plastic reorganization of neural architecture and synaptic connections, which in turn may be a sign of successful adaptation [17,72]. Thereby, sleep homeostasis is tightly linked to resilience mediated by the brain [72,73]. Animal and patient data indicate that the impaired sleep, statistically associated with neurodegeneration, is not only a result of an underlying circuit malfunction but might also directly contribute to neurodegeneration [74].

The aging process particularly challenges brain resilience [16,17,72]. Typically, age-associated molecular and behavioral changes occur gradually with aging [2,10,32,64]. In particular, cognitive functions and sleep patterns change progressively with increasing age [18,31,59,65].

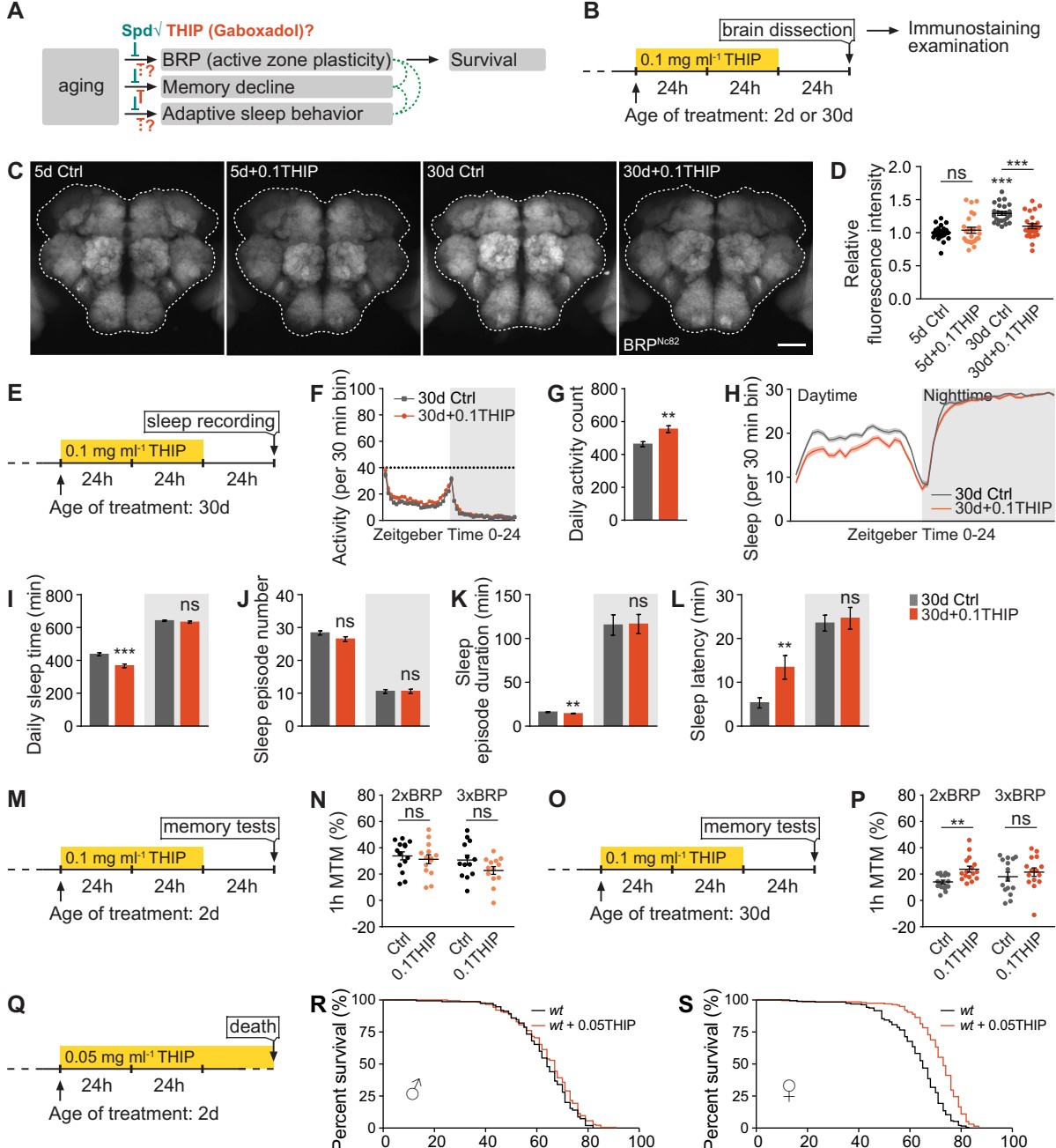

**Fig 7. Excessive deep sleep reverses age-associated active zone plasticity/PreScale and boosts lifespan. (A and B)** Rationale (**A**) and protocol (**B**) for immunochemistry examinations of fly brains of 0.1 mg ml$^{-1}$ THIP-treated *wt* female flies at ages 5d and 30d. Spd supplementation suppresses early aging-associated PreScale, memory decline, and sleep pattern changes, which might functionally intersect for survival. However, though THIP treatment suppresses age-associated memory decline, its effects on PreScale and early aging-associated sleep pattern changes were unclear. (**C and D**) Confocal images (**C**) and whole-mount brain staining analysis (**D**) of BRP in *wt* female flies at ages 5d and 30d with or without 0.1 mg ml$^{-1}$ THIP treatment. *n* = 24–25 for all groups. Scale bar: 50 μm. (**E**) Protocol for sleep test of *wt* female flies, which have been treated with 0.1 mg ml$^{-1}$ THIP for 2 days at age 30d. (**F and G**) Locomotor walking activity pattern (**F**) and statistic (**G**) of 30d *wt* flies after 2 days of 0.1 mg ml$^{-1}$ THIP treatment. (**H-L**) Sleep structure of 30d *wt* female flies after 2 days of 0.1 mg ml$^{-1}$ THIP treatment averaged from measurements over 2 days, including sleep profile plotted in 30-min bins (**H**), daytime and nighttime sleep amount (**I**), number and duration of sleep episodes (**J and K**), and sleep latencies (**L**). *n* = 90–95 for all groups. (**M-P**) Protocol (**M and O**) and 1h MTM (**N and P**) for 2xBRP and 3xBRP flies, which have been treated with 0.1 mg ml$^{-1}$ THIP for 2 days at age 2d or 30d. *n* = 13–16 for all groups. Student *t* test is shown for comparison between two groups. One-way ANOVA with Bonferroni multiple comparisons test is shown for comparisons of three groups. ** *p* < 0.01; *** *p* < 0.001; ns, not significant. Error bars: mean ± SEM. (**Q**) Protocol for longevity of 0.05 mg ml$^{-1}$ THIP-treated *wt* flies. (**R and S**) Lifespan analysis of *wt* male and female flies treated with 0.05 mg ml$^{-1}$ THIP. For male flies

(**R**), n = 176 for treated compared to untreated (n = 150, ns). For female flies (**S**), n = 199 for treated compared to untreated (n = 192, p < 0.001). Gehan–Breslow–Wilcoxon test is shown. ***p < 0.001; ns, not significant. Error bars: mean ± SEM. Underlying data can be found in S7 Data. BRP, Bruchpilot; MTM, middle-term memory; Spd, spermidine; THIP, 4,5,6,7-Tetrahydroisoxazolo[5,4-c]pyridin-3-ol; wt, wild type.

For sleep, human studies reported diverse sleep alterations in old adults, including difficulties to fall asleep, the inability to maintain sleep (insomnia), and daytime sleepiness (hypersomnia) [75]. Similarly, in flies, age-associated alterations of sleep patterns have been extensively reported but might be sensitive to the exact experimental conditions chosen [27–31,65,76]. Still, the mechanisms mediating the age-associated sleep pattern changes are poorly understood. Specifically, we still miss critical information of how aging affects neuronal and synaptic organization and plasticity, how such changes intersect with age-associated alterations of sleep pattern and cognitive function, whether neuronal and synaptic changes might be causally involved in age-associated behavioral changes and resilience, and whether critical periods for establishing resilience might exist across the aging trajectory. As a matter of fact, brain resilience might protect these entities in a concordant fashion, suggesting shared mechanisms to be at play, while differentiating truly causative from only correlative or even adaptive and protective changes is a fundamental challenge.

Our previous studies described a shared molecular signature at central *Drosophila* brain synapses upon aging [8] or sleep loss [33], which we here collectively refer to as PreScale. In this study, we investigated the dynamic range and temporal profile of PreScale across the fly lifespan and explored its consequences for cognitive and organismal fitness and longevity. We provide evidence suggesting that this form of broadly distributed presynaptic plasticity provides a resilience module, which forms on demand and allows the brain to cope with the consequences of early aging and sleep deprivation. At the same time, PreScale might acutely prioritize tasks and establish trade-offs between survival and longevity on the one and memory formation on the other hand (Fig 8). These conclusions are based on the following new findings.

Firstly, PreScale normally builds up with age to peak at mid-age, but at later advanced age, it declines again (Fig 1). Aging and sleep loss as two important stressors seemingly converge to trigger an at least very similar scenario of presynaptic active zone plasticity here. Notably, PreScale was reverted when mid-aged animals were pharmacologically triggered with extensive sleep for 2 days (Fig 7B–7D) and also does not form in mid-aged animals when supplied with Spd [8].

Secondly, concerning resilience, when genetically mimicked to a moderate degree (3xBRP), PreScale promoted organismal longevity (Fig 3A–3E). It is certainly possible that chronic genetically induced PreScale (via BRP copy number titration) might, in some regards, differ from the presynaptic plasticity observed in either acutely sleep-deprived animals or flies of mid-age. Still, direct comparisons indeed support that our experimental BRP titration can serve as a surrogate of early aging concerning behavioral alterations (Figs 2–7). Importantly, 3xBRP efficiently promoted longevity in flies chronically challenged by sleep loss (Fig 3F and 3H) but proved inefficient or even counterproductive in neurodegenerative conditions of other kind (Fig 3J–3M). In contrast, genetically triggering very high, likely unphysiological BRP levels (4xBRP) robustly shortened lifespan (Figs 3A–3C and S3L–S3V), arguing in direction of a "sweet spot" concerning its resilience-promoting function. Consistent with the dynamic range of PreScale over the aging trajectory (Figs 1 and S1), the critical period in relation to longevity promotion and memory restoration might indeed be the early aging period until mid-age, while flies of advanced age are seemingly no longer plastic and nonreversible in essential organismal functions including memory formation, at least under standard rearing conditions (Fig 6Q and 6R). Consistently, a recent study showed that rapamycin treatment in

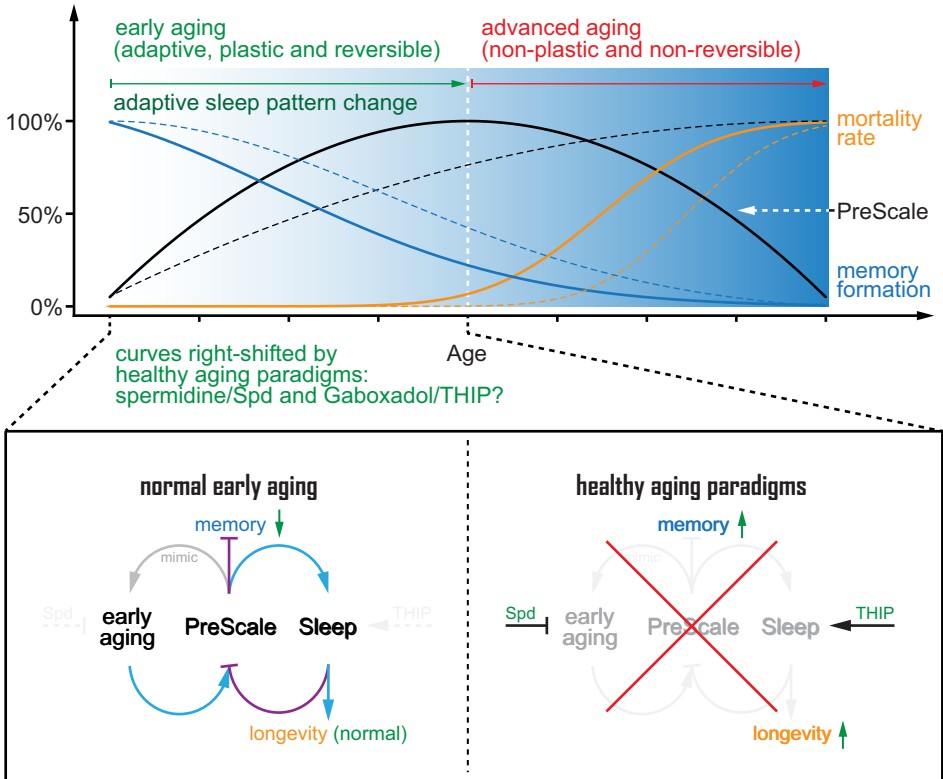

**Fig 8. A model for presynaptic active zone plasticity (PreScale) in executing trade-offs between memory and longevity.** A model for the core AZ scaffold protein BRP-driven PreScale in executing trade-offs between sleep homeostasis, memory formation, and longevity. Early aging provokes PreScale-type plasticity and adaptive sleep pattern changes and subsequently steers trade-offs between memory formation and longevity, which can be mimicked by titrating the gene copies of BRP. Rejuvenation paradigms like spermidine supplementation and Gaboxadol/THIP treatment during early aging eliminate the need of PreScale for regulating adaptive sleep patterns to steer trade-offs between memory and longevity. As a consequence, these paradigms suppress PreScale to allow for new memory formation and lifespan extension. Thus, PreScale likely executes behavioral adaptations and trade-offs during a still plastic phase of early brain aging, illustrating how life strategy manifests on a circuit and synaptic plasticity level. AZ, active zone; BRP, Bruchpilot; Spd, spermidine.

early adulthood suffices to extend lifespan but proves inefficient when applied at advanced age [77]. As we found BRP to promote sleep and regulate locomotion patterns in this early aging period (Figs 4 and S4), PreScale-driven processes might well be causally related to organismal resilience and extended longevity (Fig 8).

Thirdly, moderate BRP increases (3xBRP) tended to slightly reduce learning and memory capacity and block the beneficial effects of THIP treatment in the early aging period (Figs 3N–3Q and 7M–7P). Notably, high (4xBRP) levels even more severely reduce olfactory learning and memory [8]. The increased energy investment to support formation of long-term memories operates in a trade-off relation to save energy for better survival under food shortage or at advanced age [61,62]. Similarly, early aging-associated sleep pattern changes, for example, daytime sleepiness, could favor lower levels of energy consumption and consequently promote survival. Thus, PreScale might steer the trade-offs between the acute needs to entertain costly behaviors and/or to support sheer organismal fitness and longevity.

It might be asked why our artificial increase of *brp* gene dose from *wt* 2xBRP to 3xBRP happens to be advantageous concerning longevity under our rearing conditions. Interestingly, learning ability can be substantially improved by artificial selection in animals ranging from

*Drosophila* to rats [78], while *Drosophila* has been shown to make use of learned cues when tested in seminatural environments [79]. Along those lines, it will be very interesting to measure PreScale across the lifespan under different culture conditions, for example, aging at lower temperatures (i.e., 18˚C compared to 25˚C) was shown to extend lifespan and memory formation period [32].

PreScale might be a molecular/physiological manifestation of life strategy decisions balancing principal life history–related trade-offs. However, how would PreScale possible execute trade-offs between memory and longevity, given that PreScale itself is sleep-promoting? As healthy sleep supports subsequent learning, it might be possible that genetic installation of PreScale promotes a kind of sleep, which does not support all functional aspects of memory-supporting sleep, leading to specific lifespan extension without benefiting cognitive functions. Considering the recently reported different sleep stages in flies [80,81] and the sleep-supporting slow wave activities in the R5 network [50], PreScale-encoded sleep need might define certain sleep stages that specifically promote lifespan but not memory formation. Indeed, we recently showed that the STM impairment of 4xBRP animals was fully rescued when BRP was eliminated specifically in the R5 neurons, while sleep was only partially suppressed [33]. Thus, it is possible that such a manipulation allowed for a rebalance of the trade-offs, which might reopen the window for executing the memory function of PreScale-encoded sleep, providing a plausible alternative explanation for early aging-associated memory impairment.

From our collective data, both Spd supplementation as well as triggering excessive sleep by THIP treatment allowed to attenuate PreScale in mid-aged flies (Figs 7A–7D and 8) [8]. We speculate that some vital physiological cellular regulations might be no longer sufficiently plastic but instead steer into a detrimental direction at advanced age. Notably, Spd supplementation protects both mitochondrial respiration and autophagic flux from age-associated functional decay [68,69]. Efficient mitochondrial electron transport and autophagy in turn are coupled to sleep regulation [82,83]. Moreover, mitochondrial functionality bidirectionally regulates early aging-associated PreScale-type plasticity revealed by BRP levels [35], and autophagy defects trigger PreScale in a non-cell-autonomous manner [84]. Thus, we speculate that Spd might delay the onset of early aging-associated sleep pattern changes and memory decline through its improvements of mitochondrial functions and autophagy, while making PreScale dispensable. Acute Gaboxadol/THIP treatment might equally reset the aging brain by reversing the early aging-associated decline in mitochondrial and autophagic functions [83]. It should be warranting to further dissect the molecular mechanisms that mediate the coupling between mitochondrial and autophagic status and the trade-offs of PreScale with early aging-associated sleep pattern changes and memory decline.

Concerning the mechanistic action of PreScale on a synaptic and neurophysiological level, we observed a reprogramming of neuronal activity patterns of R5 neurons (S2A–S2E Fig) [33] and dFB neurons (Fig 2C and 2D), and the firing pattern and membrane excitability of the dFB neurons were changed upon genetic installation of PreScale and early aging (Fig 2E–2P). Interestingly, both reducing and decreasing *brp* gene copies suppressed the activity of dFB neurons, indicated by the $Ca^{2+}$-dependent activity reporter CaLexA [48], which is meant to be a proxy for the activity history of neurons (Figs 2C, 2D and S2F–S2J). Taken together, we reasoned that the reduced dFB CaLexA signal in 1xBRP animals likely represents reduced and less consolidated sleep, while in 3xBRP and 4xBRP animals, it mimics a sleep-deprived state [33]. Alternatively, because dFB and R5 neurons receive different information input in the brain network, for example, dFB neurons receive inputs from wake-promoting dopaminergic neurons [53]. In addition, a circadian output circuit innervates R5 neurons to regulate sleep [85,86]. Thus, the different activity states of dFB and R5 neurons might represent a complex physiological state that integrates different input information to coordinate behaviors.

Notably, a recent study suggests that dFB neurons promote a specific form of sleep with brain activity that is different from spontaneous sleep [81]. As PreScale can per se change short-term plasticity and the transmission efficacy in a frequency-dependent manner [33], we speculate that this form of synapse remodeling might promote the transmission of specific activity patterns. Favoring transmission in certain frequency domains might also explain its sleep-promoting action, as previous work identified slow wave oscillatory activities to be associated with sleep regulation via ellipsoid body R5 neurons [50]. In addition to R5 neurons, the interconnected dFB neurons have been shown to exhibit oscillatory activity and regulate both sleep and memory [20,70,71,80,81]. It is tempting to speculate that dFB neurons might be at the intersection of PreScale-mediated trade-offs between longevity and cognitive functions, and the reduced neuronal activity and membrane excitability might explain the impaired memory in 3xBRP and 4xBRP animals. Potentially in the dFB/R5 network, skewing the representation of sensory information might also explain the reduced proclivity of flies under PreScale for forming new memories (Fig 8). How exactly the distinct effects of PreScale integrate from synapse over circuit and neuronal activity to behavioral level remains a warranting question for future analysis.

## Materials and methods

### Fly genetics and maintenance

$w^{1118}$ (*iso31*, BDSC#5905) was used as *wt*, 2xBRP, and background control. A *brp* null mutation (*c04298*, BDSC#85966) was utilized to reduce *brp* gene copy [33,40]. A genomic *brp* P [acman] construct [87], which was mapped to be integrated into the 5′ UTR of *CG11357* (S3L and S3M Fig) [88], was used to increase *brp* gene copy number from 2xBRP to 3xBRP and 4xBRP [33] or to rescue the sleep phenotypes of 1xBRP flies (gBRP). A line with a P-element inserted at the 5′ UTR of *CG11357* (*EY12484*, BDSC#20838) was acquired to mimic the effects of a P-element at *CG11357* 5′ UTR, which was largely comparable to *wt* (S3N–S3V Fig). Short sleep mutants *wake* (*EY02219*, BDSC#15858) and *inc* (*f00285*, BDSC#18307) and autophagy *atg7* mutant (*d06996*, BDSC#19257) were described previously [56,57,89]. Alzheimer's disease model flies were generated by pan-neuronal *elav-Gal4*-driven expression of *Aβ-arctic* (BDSC#458 and #33774). *UAS-mCD8.GFP* (BDSC#32188) was expressed by *elav-Gal4* as control for Alzheimer's disease model flies or by *R23E10-Gal4* for patch-clamp electrophysiology. CaLexA (BDSC#66542) was expressed pan-neuronally by *elav-Gal4* or in either dFB neurons marked by *R23E10-Gal4* [53] or R5 neurons driven by *R58H05-Gal4* [51]. All fly strains were backcrossed to $w^{1118}$ *wt* background for at least six generations to remove potential genetic modifiers.

Flies were raised under standard laboratory conditions on semidefined medium (Bloomington recipe) under 12/12h light/dark cycles with 65% humidity at 25°C. Unless specifically stated, female flies at certain ages were used except aversive olfactory memory experiments in which mixed populations of both sexes were used.

### Aging and longevity

During the development at larval stages, the density of larvae was purposely controlled across groups and genotypes. Aging was performed with mixed populations of male and female flies. To avoid stressing the flies during aging (for example, flies step on each other), the amount of flies within each individual vials was grossly controlled across groups, genotypes, and experiments. The aging flies were routinely flipped onto fresh food every second day or over the weekends until a desired age was reached for experiments.

Longevity experiments were carried out similarly, except that female and male flies were separated and sorted into a population of approximately 20 to 25 flies per vial/replicate at age 2d after fully mating. To reduce the variability in lifespan, a few different cohorts of flies were used for each experiment. Similar to aging flies, the longevity flies were regularly transferred onto fresh food and the amount of dead flies in each vial was recorded at each time of transfer until the death of the last fly of a replicate.

## Western blotting

Western blot analysis was carried out as previously reported [33]. Basically, female flies at specific ages were dissected in ice-cold Ringer's solution (pH = 7.3, mOsm = 290 to 310, with 5 mM HEPES-NaOH, 130 mM NaCl, 5 mM KCl, 2 mM $MgCl_2$, 2 mM $CaCl_2$, and 36 mM sucrose) between ZT6 and ZT10 (ZT, zeitgeber time). The brain samples were stored at −20°C for short term (approximately 2 weeks) or at −80°C for longer. Brain samples of the same experiment were fully homogenized by three rounds of intense vortexing in lysis buffer (0.5% Triton X-100, 2% SDS, 1× protease inhibitor, 1× sample buffer in PBS) followed by full-speed centrifugation for 6 min at 18°C. The samples were then intensively vortexed to resuspend the brain tissue and centrifuged again for 6 min at 18°C. To avoid any saturation in the amount of proteins for detecting the differences, less than one brain's supernatant was loaded into each lane for SDS-PAGE and immunoblotting. Thereafter, the blots were manually developed with ECL solutions and Kodak/GE films. The following primary antibodies were used: mouse anti-BRP Nc82 (1:1,000 to 1:2,000), rabbit anti-BRP D2 (rb5228, 1:100,000), guinea pig anti-Unc13A (14GP17, 1:2,000), mouse anti-Dlg1 (4F3, 1:3,000), mouse anti-Syn (3C11, 1:2,000), mouse anti-Tubulin (Sigma T9026, 1:100,000), mouse anti-Syx (8C3, 1:2,000), and rabbit anti-Atg8a (Ab109364, 1:1,000). The developed films without saturation were scanned by an EPSON V330 scanner in 16-bit grayscale tiff format [33].

## Immunostaining

Immunostaining was performed exactly as previously described [33]. Adult female flies at certain ages, after sleep deprivation, or after THIP treatment were dissected in ice-cold Ringer's solution and immediately fixed in 4% paraformaldehyde (pH = approximately 7.3) for 30 min at room temperature. After fixing, brains were washed in 0.7% PBST (PBS with 0.7% Triton X-100, v/v) for 3 or 4 times for a total of 1 h and blocked in 0.7% PBST with 10% normal goat serum (v/v) for at least 2 h at room temperature. Primary antibodies were diluted in 0.7% PBST with 5% NGS for primary antibody incubation at 4°C. Afterwards, brains were washed again in 0.7% PBST for at least 4 times and then incubated with secondary antibodies diluted in 0.7% PBST with 5% NGS in darkness. Finally, after secondary antibody incubation, brains were washed for at least 4 times and mounted in Vectashield and stored at 4°C in darkness for confocal microscopy. Primary and secondary antibodies were incubated over a night, except for BRP Nc82 primary antibody, which requires at least two nights' incubation for proper staining quality. The following primary antibodies were used: mouse anti-BRP Nc82 (1:50) and chicken anti-GFP (ab13970, 1:1,500). Goat anti-rabbit Cy5 and Alexa Fluor 647, goat anti-mouse Alexa 488, and goat anti-chicken Alexa 488 were diluted at 1:300 for secondary antibody incubation.

## Confocal microscope image acquisition, processing, and analysis

Whole-mount brain samples were scanned with a Leica TCS SP8 confocal microscope under oil objectives. All the parameters of the microscope were set to avoid saturation and kept constant throughout each experiment for the purpose of comparing fluorescence intensity. All

images (confocal and western blot images) were processed and analyzed in ImageJ (Fiji) software (https://fiji.sc/). For stacks of brain images, an average intensity projection was chosen and the intensity of the single projected image was analyzed by manually drawing a region of interest to determine the gray value within this ROI. The values for each replicate were normalized to *wt* or control, and different replicates were pooled after normalization.

## Sleep and sleep deprivation

Sleep and sleep deprivation experiments were performed as previously published [33]. Briefly, locomotor walking activity and sleep were recorded by *Drosophila* Activity Monitors (DAM2) from Trikinetics (Waltham, MA) in 12/12-h light/dark cycles at 25°C with 65% humidity. Single flies at specific age were individually housed in Trikinetics glass tubes (5 mm inner diameter and 65 mm length), which had 5% sucrose and 2% agar in one side of the tube. The locomotor walking activity was uploaded every 1 min, and the data from the first day was excluded due to the entrainment to new environments. A period of immobility without locomotor walking activity lasting for at least 5 min was defined as sleep [24]. Sleep and activity were analyzed using the Sleep and Circadian Analysis MATLAB Program [90].

Sleep deprivation was performed exactly as previously described [33]. The DAM2 monitors were fixed onto a Vortexer Mounting Plate (Trikinetics) on an Analog Multi-Tube Vortexer controlled by a Trikinetics LC4 light controller and an acquisition software for sleep deprivation configuration. Pulses of vortex lasting for 1.2 s were applied randomly with interpulse intervals between 0 s and 40 s to fully sleep deprive flies during night from ZT12 to ZT24. As female flies show strong and consistent early aging-associated alterations in sleep patterns, females were used for all sleep experiments, except for S3A–S3K Fig, within which male or female flies were used.

## Aversive olfactory memory

Associative aversive olfactory memory experiments were performed as previously described [8,33]. Two aversive odors, 3-octanol (Oct) (1:100 dilution) and 4-methylcyclohexanol (MCH) (1:100 dilution), were used as olfactory cues (odors were diluted in mineral oil) and 120 V AC current electrical shocks served as a behavioral reinforcer. Briefly, during training, about 100 flies were sequentially exposed to the first odor (conditioned stimulus, CS+, MCH or Oct) paired with electrical shocks (unconditioned stimulus, US) for 60 s followed by 60 s rest and then the second odor (CS−, Oct or MCH) without electrical shocks for 60 s. During testing, these flies were exposed simultaneously to both odors (CS+ and CS−) and had 60 s to choose between the two odors. A reciprocal experiment in which the other of the two odors was paired with electrical shocks was performed simultaneously. A memory index was calculated as the number of flies that chose the CS− odor minus the number of flies that chose the CS+ odor divided by the total number of flies. The final memory index was averaged from the two memory indices from the reciprocal experiments.

STM was tested immediately after training, while 1h MTM and 3h MTM were tested 1 h or 3 h after training. For 1h ARM, one of the two 1h MTM components, flies received cold shock on ice for 90 s, 30 min after training. 1h ARM was measured 30 min after cold shock, and ASM was calculated by subtracting ARM from MTM.

Odor avoidance experiments were carried out similarly. Basically, naive flies were exposed to only one of the two odors and were allowed to choose between this odor and the air. The performance index was calculated as the number of flies that chose the air minus the number of flies that chose the odor divided by the total number of flies.

## Spermidine supplementation and Gaboxadol/THIP treatment

For acute THIP (Sigma-Aldrich, Cat# T101) treatment for memory tests, adult flies at certain ages were transferred onto normal fly food with specific concentrations of THIP at approximately ZT1 for either 48 h or 72 h. Flies were either tested immediately after treatment or flipped back to normal fly food without THIP and tested 24 h after THIP treatment. For acute THIP treatment for sleep experiments, specific concentrations of THIP were contained in 5% sucrose and 2% agar medium, and sleep was measured as described above. For acute THIP treatment for immunostaining, flies were treated with 0.1 mg ml$^{-1}$ THIP for 48 h and dissected 24 h after treatment.

Spd (Sigma-Aldrich, Cat# S2626) treatment has been described previously [8,10]. Flies were raised and aged to specific days on 5 mM Spd containing normal fly food for sleep experiments with 5% sucrose and 2% agar medium.

## In vivo patch-clamp

Five- and 20-day-old female flies (*R23E10>mCD8-GFP*) were used for in vivo patch-clamp electrophysiology recording of dFB neurons. Adult *Drosophila* preparation for in vivo electrophysiology recording was performed based on a former study [91]. Flies were anesthetized by keeping them on ice for 1 to 2 min, and single flies were then fixed to the chamber with paraffin wax and dissected in external solution (103 mM NaCl, 3 mM KCl, 1.5 mM CaCl$_2$, 4 mM MgCl$_2$, 1 mM NaH$_2$PO$_4$, 26 mM NaHCO$_3$, 5 mM TES (N-tris[hydroxymethyl]methyl-2-aminoethane sulfonic acid), 8 mM Trehalose, 10 mM Glucose, and 7 mM Sucrose, Osm = 280 ± 3) preequilibrated with carbogen (95% O$_2$/5% CO$_2$). The head cuticle on the posterior surface was peeled off to reveal the dFB neurons somata, and the glial sheath around the targeted area was focally removed with sharpened forceps. GFP-labeled dFB neurons were visualized under a 40 × objective. Signals were captured with a CCD digital camera (HAMAMATSU, ORCA-ER) mounted on the microscope.

Recording from dFB neurons was performed as described previously [53]. During recording, the preparation was superfused continuously with the external solution. Somata of dFB neurons were targeted with pipettes (8 to 10 MΩ) filled with internal solution (140 mM K-aspartate, 1 mM KCl, 10 mM HEPES, 1 mM EGTA, 4 mM MgATP, and 0.5 mM Na$_3$GTP, pH = 7.3 and Osm = 265). Recordings were acquired with a MultiClamp 700B amplifier (Molecular Devices) and sampled with a Digidata 1440A interface (Molecular Devices). Signals were filtered at 6 kHz and digitized at 10 to 20 kHz. Data were analyzed using Clampfit 10.7 (Molecular Devices).

To measure the input resistance, small steps of 1-s hyperpolarizing current pulses were injected into the patched cell. To estimate the excitability of dFB neurons, cells were pre-hold at −60 mV followed by injection of a series of current steps (from −10 to 100 pA, 5 pA increment per step). Liquid junction potential of 13 mV was corrected offline. Spikes were searched with the event detection function in Clampfit. The following criteria were used for bursts identification: interspike interval <80 ms, number of spikes >4. For comparisons of interspike interval distribution, Kolmogorov–Smirnov tests were performed. Cells with an access resistance higher than 50 MΩ were excluded from the analysis.

## Statistics

GraphPad Prism 7 was used to create the figures and perform most statistics. The Student *t* test was used for comparison between two groups, and one-way ANOVA with Bonferroni multiple comparisons test was used for multiple comparisons between multiple groups (≥3). Two-way repeated-measures ANOVA was used for sleep rebound experiments. Longevity

data was analyzed by Gehan–Breslow–Wilcoxon test in order to put more weight on earlier time points to compare differences between survival curves. Asterisks or ns (not significant) above a group indicate the comparison of this group to 2xBRP *wt* or controls; asterisks or ns above a line denote the comparison between the two specific groups for most statistical analyses, unless stated otherwise.

## Supporting information

**S1 Fig. Synaptic plasticity with aging.** (**A**-**E**) Representative western blots (**A**) and statistics of a spectrum of synaptic proteins, including BRP Nc82 (**B**), Dlg1 (**C**), Syn (**D**), and Syx (**E**) in *wt* female flies with aging. $n = 3$. One-way ANOVA with Bonferroni multiple comparisons test is shown. $^*p < 0.05$; $^{**}p < 0.01$; $^{***}p < 0.001$; ns, not significant. Error bars: mean ± SEM. Underlying data can be found in S1 Data Sheet. Raw images of this figure are provided in S1 Raw Images. Dlg1, Discs large; Syn, Synapsin; Syx, Syntaxin; *wt*, wild type.
(TIF)

**S2 Fig. PreScale plasticity provokes activity reprogramming in R5 and dFB neurons.** (**A**) CaLexA signal is likely the representation of activity history of neurons. (**B** and **C**) Confocal images (**B**) and whole-mount brain staining analysis (**C**) of CaLexA signal intensity with CaLexA expressed in R5 neurons by *R58H05-Gal4* in 4xBRP compared to 2xBRP flies. $n = 20$ for all groups. (**D** and **E**) Confocal images (**D**) and whole-mount brain staining analysis (**E**) of CaLexA signal intensity with CaLexA expressed in R5 neurons by *R58H05-Gal4* in 1xBRP compared to 2xBRP flies. $n = 12$ for all groups. (**F**) Scheme of an interconnected sleep circuit in the central complex composed of the dFB (marked by *R23E10-Gal4*), Helicon cells, and the ellipsoid body R5 neurons (R5 marked by *R58H05-Gal4*). R5 neurons also receive circadian inputs. (**G** and **H**) Confocal images (**G**) and whole-mount brain staining analysis (**H**) of CaLexA signal intensity with CaLexA expressed in R5 neurons by *R23E10-Gal4* in 1xBRP compared to 2xBRP flies. $n = 10–12$. (**I** and **J**) Confocal images (**I**) and whole-mount brain staining analysis (**J**) of CaLexA signal intensity with CaLexA expressed in R5 neurons by *R23E10-Gal4* in 1xBRP compared to 2xBRP flies, and in sleep-deprived flies. Sleep deprivation was performed between ZT12 to ZT24 during nighttime. $n = 12$ for all groups. Student *t* test is shown. $^*p < 0.05$; $^{**}p < 0.01$; $^{***}p < 0.001$; ns, not significant. Scale bar: 20 μm. Error bars: mean ± SEM. (**K**) Confocal image of the expression pattern of *R23E10-Gal4* indicated by GFP staining. Scale bar: 100 μm. (**L**) Widefield image of the cell bodies of *R23E10-Gal4* indicated by live GFP signal, the cell body localization is indicated in the dashed circles of (**J**). Scale bar: 10 μm. Underlying data can be found in S1 Data Sheet. dFB, dorsal fan-shaped body; ZT, zeitgeber time.
(TIF)

**S3 Fig. BRP promotes sleep in a dosage-dependent manner in male flies, similar to females.** (**A**-**E**) Sleep structure of 1xBRP-4xBRP male flies averaged from measurements over 2–4 days, including sleep profile plotted in 30-min bins (**A**), daily sleep amount (**B**), number and duration of sleep episodes (**C** and **D**), and sleep latencies (**E**). $n = 63–80$. One-way ANOVA with Bonferroni multiple comparisons test is shown. (**F**) Linear regression analysis of daily sleep amount in flies with different *brp* copies in both male ($R^2 = 0.81$) and female ($R^2 = 0.95$) flies. $n = 63–80$ per group for male, $n = 123–128$ per group for female. (**G**-**K**) Sleep structure of 5d female and male *wt* flies from measurements over 2–4 days, including sleep profile plotted in 30-min bins (**G**), daily sleep amount (**H**), number and duration of sleep episodes (**I** and **J**), and sleep latencies (**K**). $n = 62–64$ per group. (**L**) Genomic mapping and sequence of the integration site of the *brp* P[acman] transgenic construct. Red letters indicate *brp* P[acman] sequence, and black letters indicate genomic *CG11357* gene sequence. (**M**) Simplified

gene span of *CG11357* and the integration site of the *brp* P[acman] and another P-element mediated allele *EY12484* that are both localized at the 5′ UTR region of *CG11357*. (**N-R**) Sleep structure of *EY12484* female flies averaged from measurements over 2–4 days, including sleep profile plotted in 30-min bins (**N**), daily sleep amount (**O**), number and duration of sleep episodes (**P** and **Q**), and sleep latencies (**R**). $n = 55$ for *wt* control and $n = 32$ for *EY12484*. Student *t* test is shown. (**S** and **T**) An independent experiment of the lifespan analysis of 2xBRP and 4xBRP flies. For male flies (**S**), $n = 134$ for 4xBRP compared to 2xBRP ($n = 132$, $p < 0.001$). For female flies (**T**), $n = 134$ for 4xBRP compared to 2xBRP *wt* control flies ($n = 141$, $p < 0.001$). (**U** and **V**) Lifespan analysis of 2xBRP *wt* and *EY12484* flies. For male flies (**U**), $n = 128$ for *EY12484* compared to 2xBRP ($n = 127$, ns). For female flies (**V**), $n = 129$ for *EY12484* compared to 2xBRP *wt* control flies ($n = 127$, ns). Gehan–Breslow–Wilcoxon test is shown for all longevity experiments. $^{*}p < 0.05$; $^{**}p < 0.01$; $^{***}p < 0.001$; ns, not significant. Error bars: mean ± SEM. Underlying data can be found in S1 Data Sheet. BRP, Bruchpilot; CDS, coding DNA sequence; *wt*, wild type.
(TIF)

**S4 Fig. Early aging-associated sleep pattern changes of 3xBRP animals.** (**A-D**) Sleep structure of 1xBRP female flies at age 20d rescued by a transgenic *brp* copy (gBRP) and averaged from measurements over 2–3 days, including daily sleep amount (**A**), number and duration of sleep episodes (**B** and **C**), and sleep latencies (**D**). $n = 63–64$ for all groups. One-way ANOVA with Bonferroni multiple comparisons test is shown. (**E** and **F**) Locomotor walking activity distribution across the day (**E**) and averaged daily total walking activity (**F**) of 3xBRP compared to 2xBRP female flies at ages 5d, 20d, 30d, and 40d. (**G-O**) Sleep structure of 3xBRP female flies at ages 5d, 20d, 30d, and 40d averaged from measurements over 2–3 days, including sleep profile plotted in 30-min bins (**G**), daytime and nighttime sleep amount (**H** and **L**), number and duration of sleep episodes (**I, J, M,** and **N**), and sleep latencies (**K** and **O**). $n = 246–247$ for 5d, $n = 61–63$ for 20d, $n = 59–62$ for 30d, and $n = 32$ for 40d. Two-way ANOVA with Sidak multiple comparisons is shown. $^{*}p < 0.05$; $^{**}p < 0.01$; $^{***}p < 0.001$; ns, not significant. Error bars: mean ± SEM. Underlying data can be found in S1 Data Sheet.
(TIF)

**S5 Fig. Spd supplementation does not affect sleep rebound of 5d young animals.** (**A** and **B**) Rationale (**A**) and protocol (**B**) for the consequence of Spd supplementation in age-associated alterations of sleep pattern. Early aging trigger PreScale, memory decline, and sleep pattern changes, which might functionally intersect for survival. Spd supplementation was shown to suppress PreScale and memory decline, but its effect on early aging-associated sleep pattern changes was unclear. (**C**) Sleep profile for 5d *wt* female flies treated with 5 mM Spd compared to untreated for 3 consecutive days. (**D**) Normalized cumulative sleep loss during 12-h nighttime sleep deprivation and 24-h sleep rebound. Two-way repeated-measures ANOVA with Fisher LSD test did not detect any significant treatment × time interaction ($F_{(47, 2784)} = 0.0038$; $p > 0.9999$) during sleep rebound. (**E**) Sleep recovered at three different time points after sleep deprivation for 5d 5 mM Spd-treated compared to untreated female flies. $n = 29–31$ for both groups. Two-way ANOVA with Sidak multiple comparisons is shown. ns, not significant. Error bars: mean ± SEM. Underlying data can be found in S1 Data Sheet. LSD, least significant difference; Spd, spermidine; *wt*, wild type.
(TIF)

**S6 Fig. Acute deep sleep induced by Gaboxadol/THIP feeding.** (**A**) Protocol for sleep test of *wt* female flies treated with different concentrations of THIP at either age 3d or 30d. (**B-F**) Sleep structure of 5d *wt* female flies fed with 0.05 mg ml$^{-1}$ and 0.1 mg ml$^{-1}$ THIP from

measurements over 2–3 days, including sleep profile plotted in 30-min bins (**B**), daily sleep amount (**C**), number and duration of sleep episodes (**D** and **E**), and sleep latencies (**F**). $n = 64$ for untreated control *wt* flies, $n = 32$ for both 0.05 mg ml$^{-1}$ and 0.1 mg ml$^{-1}$ THIP-treated groups. One-way ANOVA with Bonferroni multiple comparisons test is shown. (**G**-**K**) Sleep structure of 30d *wt* female flies fed with 0.05 mg ml$^{-1}$ THIP from measurements over 2–3 days, including sleep profile plotted in 30-min bins (**G**), daily sleep amount (**H**), number and duration of sleep episodes (**I** and **J**), and sleep latencies (**K**). $n = 31$ for both groups. Student *t* test is shown. $^{*}p < 0.05$; $^{**}p < 0.01$; $^{***}p < 0.001$; ns, not significant. Error bars: mean ± SEM. Underlying data can be found in S1 Data Sheet.
(TIF)

**S7 Fig. Acute Gaboxadol/THIP treatment for 2 days does not alter sleep behavior 1 day after the treatment in young animals.** (**A**) Protocol for sleep test of *wt* female flies that have been treated with 0.1 mg ml$^{-1}$ THIP for 2 days at age 2d. (**B** and **C**) Locomotor walking activity pattern (**B**) and statistic (**C**) of 30d *wt* female flies after 2 days of 0.1 mg ml$^{-1}$ THIP treatment. (**D**-**H**) Sleep structure of 2d *wt* female flies after 2 days of 0.1 mg ml$^{-1}$ THIP treatment averaged from measurements over 2 days, including sleep profile plotted in 30-min bins (**D**), daytime and nighttime sleep amount (**E**), number and duration of sleep episodes (**F** and **G**), and sleep latencies (**H**). $n = 32$ for all groups. Student *t* test is shown. ns, not significant. Error bars: mean ± SEM. Underlying data can be found in S1 Data Sheet.
(TIF)

**S1 Data. Data underlying Fig 1.**
(XLSX)

**S2 Data. Data underlying Fig 2.**
(XLSX)

**S3 Data. Data underlying Fig 3.**
(XLSX)

**S4 Data. Data underlying Fig 4.**
(XLSX)

**S5 Data. Data underlying Fig 5.**
(XLSX)

**S6 Data. Data underlying Fig 6.**
(XLSX)

**S7 Data. Data underlying Fig 7.**
(XLSX)

**S1 Data Sheet. Data underlying S1–S7 Figs.**
(XLSX)

**S1 Raw Images. Original uncropped immunoblots for Figs 1 and S1.**
(PDF)

## Acknowledgments

We thank S. Yildirim-Brochno, D. Wachmann, and Y. Ni for technical support; the Bloomington Stock Center (BDSC) for fly lines; and the Developmental Studies Hybridoma Bank (DSHB) for antibodies.

## Author Contributions

**Conceptualization:** Sheng Huang, Stephan J. Sigrist.

**Data curation:** Sheng Huang, Chengji Piao.

**Formal analysis:** Sheng Huang, Chengji Piao, Christine B. Beuschel, Zhiying Zhao.

**Funding acquisition:** Stephan J. Sigrist.

**Investigation:** Sheng Huang, Chengji Piao, Christine B. Beuschel, Zhiying Zhao.

**Methodology:** Sheng Huang, Chengji Piao.

**Project administration:** Stephan J. Sigrist.

**Supervision:** Stephan J. Sigrist.

**Writing – original draft:** Sheng Huang, Stephan J. Sigrist.

**Writing – review & editing:** Sheng Huang.

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
