## [Editor Report · Decision Letter 0]

29 Jun 2022

Dear Dr Sigrist, 

Thank you for submitting your manuscript entitled "A brain-wide form of presynaptic active zone plasticity orchestrates resilience to brain aging in Drosophila" for consideration as a Research Article by PLOS Biology.

Your manuscript has now been evaluated by the PLOS Biology editorial staff as well as by an academic editor with relevant expertise, and I am writing to let you know that we would like to send your submission out for external peer review. While we are, in principle, interested in the study, I should note that we have yet to make a firm call about whether the findings reported here go far enough beyond other work on this topic, and so we will be looking for enthusiasm amongst the reviewers towards the overall advance of the study and its fit for PLOS Biology.

Before we can send your manuscript to reviewers, we need you to complete your submission by providing the metadata that is required for full assessment. To this end, please login to Editorial Manager where you will find the paper in the 'Submissions Needing Revisions' folder on your homepage. Please click 'Revise Submission' from the Action Links and complete all additional questions in the submission questionnaire.

Once your full submission is complete, your paper will undergo a series of checks in preparation for peer review. After your manuscript has passed the checks it will be sent out for review. To provide the metadata for your submission, please Login to Editorial Manager (https://www.editorialmanager.com/pbiology) within two working days, i.e. by Jul 01 2022 11:59PM.

Kind regards,

Luke

Lucas Smith, Ph.D.

Associate Editor

PLOS Biology

lsmith@plos.org

---

## [Decision Letter · Decision Letter 1]

28 Jul 2022

Dear Dr Sigrist,

Thank you for your patience while your manuscript "A brain-wide form of presynaptic active zone plasticity orchestrates resilience to brain aging in Drosophila" was peer-reviewed at PLOS Biology. It has now been evaluated by the PLOS Biology editors, an Academic Editor with relevant expertise, and by several independent reviewers. 

The reviews of your manuscript are appended below. As you will see, while the reviewers have commented that the study is potentially interesting, they have also raised a number of important and overlapping concerns which would need to be thoroughly addressed before we can consider your manuscript further for publication. The reviewers have highlighted that currently, the manuscript is disorganized and the writing is unclear, to the extent that it is difficult to judge the advance of your findings. Additionally, the reviewers have suggested additional data and analyses be provided to strengthen the study.

In light of these concerns, we do not feel able to accept your manuscript in its current form, but we would welcome resubmission of a much revised manuscript that addresses the reviewer comments. We would expect the revised manuscript to provide additional data, and to carefully re-write the manuscript to make it more accessible to a broad audience, clarify the model, and to better explain the advance of the study.

Given the extent of revision needed, we cannot make a decision about publication until we have seen the revised manuscript and your response to the reviewers' comments. Your revised manuscript is likely to be sent for further evaluation by all or a subset of the reviewers. I should note that it is possible that, after rewriting the manuscript to improve its clarity, the reviewers may identify new concerns that were not apparent in the current version and we will need to take those into account.

**IMPORTANT - SUBMITTING YOUR REVISION**

*Re-submission Checklist*

*Published Peer Review*

*PLOS Data Policy*

*Blot and Gel Data Policy*

Sincerely,

Lucas

Lucas Smith, Ph.D.

Associate Editor

PLOS Biology

lsmith@plos.org

REVIEWS:

Reviewer #1: Huang et al. characterize a presynaptic phenomenon called PreScale and propose that age-dependent induction of PreScale may be responsible for regulating lifespan and age-related changes in sleep and memory. Overall, their manuscript is somewhat disorganized, the presentation of their data seems haphazard, and the description of their model is difficult to understand. Unfortunately, the amount of progress in this work compared to previously published work from this group is unclear. 

Specific comments:

1) The Introduction is short and vague. The first two paragraphs, which make up the bulk of the Introduction, are general background, and the important third paragraph, which introduces PreScale and spermidine, is very brief, making it difficult to understand what PreScale is and why it is important. Because the Introduction is short, much introductory material is inserted into the Results section making the logic of the manuscript more difficult to follow. Without first reading Gupta et al., 2016 and Huang et al., 2020, many readers may have difficulty understanding the rationale behind most of the experiments in the current manuscript. In fact, the authors reference a supplementary figure in Huang et al., 2020, requiring readers to look up this figure. A better introduction to these papers may improve the organization and apparent significance of this manuscript.

2) The model presented by the authors is vague, and the interactions between sleep, lifespan, memory and PreScale are unclear. For example, in the graphic abstract and in Figure S7, both spermidine and gaboxadol decrease PreScale, improve memory and extend lifespan, but they have opposite effects on sleep. Do the authors think that PreScale and sleep are unrelated? This seems unlikely since this group published Huang et al., 2020. The partial models presented in Figures S5A and 7A are difficult to understand. They seem to suggest that if PreScale is inhibited, sleep is reduced. On the other hand, if sleep is increased, PreScale is inhibited. These seem to be opposing effects that the authors could discuss more clearly.

3) Many of the results in this manuscript are correlative, and causative relationships are unclear. For example, both spermidine and gaboxadol inhibit PreScale. However, they have opposing effects on sleep, suggesting that at least for one of these interventions, effects on sleep are not mediated by PreScale. Since sleep is known to be linked to memory and lifespan, it is unclear whether the effects of eg. gaboxadol on lifespan and memory are caused by changes in PreScale or by changes in sleep. Epistatic experiments addressing causation were not performed.

4) Did the authors examine the effects of age-dependent induction of bruchpilot RNAi on sleep, lifespan, and memory? Does this increase age-dependent lifespan and memory and maintain young sleep amounts? 

5) Did the authors examine the effects of 1xBRP on dFB and R5? Are these effects the opposite of 3xBRP? 

6) How spontaneous activity and membrane excitability of dFB neurons are affected by spermidine and gaboxadol?

Reviewer #2: Huang et al. examined age-associated changes in synaptic protein levels and neuronal excitability, as well as sleep, memory, and longevity of flies with genetically manipulated BRP levels or those treated with spermidine or THIP. The results are potentially interesting, but I have several major concerns regarding the conceptual framework, data interpretation, and experimental design. A primary concern is that they do not explore sex differences in their data and do not examine reproductive fitness as a critical factor in a fly's life. Moreover, the authors conclude that "a brain-wide form of presynaptic active zone plasticity ("PreScale") promotes resilience by coupling sleep, longevity and memory during aging (abstract)." However, while the manuscript provides correlative data, it is unclear whether they point to a causal role of PreScale in the coordinated age-associated changes. Specific concerns are listed below. 

1. In what sense is the tradeoff between longevity and memory during early aging optimal? What is optimized? It seems that simply living longer wouldn't be the core purpose of a fly. One could argue that producing lots of healthy progeny would be the fly's core mission. The authors should examine reproductive fitness (e.g., # of offspring) as a critical aspect of the necessary tradeoffs as flies age. 

2. The data show substantial sex differences (e.g., Fig. 3), but there is no serious discussion of the subject. As mentioned in Point #1, the authors should examine how reproductive success is impacted by various genetic and drug manipulations.

3. The authors propose that PreScale promotes resilience during early aging by favoring longevity over memory. The finding that 3xBRP promotes longevity at the expense of memory seems consistent with the view. However, Spd and THIP promote both memory and longevity while suppressing PreScale. If Prescale has adaptive advantages, why do spermidine and THIP, which suppress Prescale, improve both memory and longevity? The authors suggest that Spd and THIP treatment promote efficient mitochondrial electron transport and autophagy, making PreScale unnecessary. How can we tell a priori whether a particular PreScale manipulation will enhance both memory and longevity or favor one over the other? What is meant by "a specific, context-dependent role of PreScale in regulating lifespan (Line 182)." Please elaborate on the specific contexts. If wake and inc mutants already have increased BRP, why does it help to have even more? 

4. The increased longevity due to PreScale and THIP treatment could be because they sleep more and therefore spend less energy. The disadvantage may be that they do not have the opportunity to produce many offspring. Again, if the authors consider reproductive fitness, the increased longevity accompanied by increased sleep may not be a good tradeoff. 

5. Fig. 2 shows striking physiological effects of increasing BRP levels to 4x. However, sleep and longevity data suggest that whereas 3xBRP provides some beneficial effects similar to PreScale, 4xBRP is detrimental. Given this finding, they should examine the electrophysiological properties for 3xBRP, not 4xBRP.

6. Why does PreScale have opposite effects on the neuronal excitability of R5 vs. dFB?

7. In the model (graphical abstract and Fig. S7), they show "longevity" as something affected by aging. Since longevity stays constant across the lifespan, mortality, which increases with aging, would be more helpful. 

8. The effects of THIP on memory seem variable across experiments. For example, 0.1THIP had no effect on STM in one experiment (H) but a significant effect in another (N). The effects of 3xBRP on longevity are also variable across experiments (Fig. 3C vs. E). What accounts for the differences?

9. Why do synaptic protein levels decline after middle age? Do older flies remember better after PreScale returns to the young fly level?

10. Some figure legends say "flies." Please specify whether they are male or female flies.

---

## [Editor Report · Decision Letter 2]

1 Nov 2022

Dear Dr Sigrist,

Thank you for your patience while we considered your revised manuscript "A brain-wide form of presynaptic active zone plasticity orchestrates resilience to brain aging in Drosophila" for publication as a Research Article at PLOS Biology. This revised version of your manuscript has been evaluated by the PLOS Biology editors and by the Academic Editor.

Overall, the Academic Editor appreciates the amount of work that has gone into this revision and is satisfied by the changes made in response to the reviewers, and so we are likely to accept this manuscript for publication. However before we can editorially accept your study, we need you to address the following data and other policy-related requests.

**IMPORTANT: Please attend to the following editorial requests in a revised manuscript: 

We require the original, uncropped and minimally adjusted images supporting all blot and gel results reported in an article's figures or Supporting Information files. We will require these files before a manuscript can be accepted so please prepare and upload them now. Please carefully read our guidelines for how to prepare and upload this data: https://journals.plos.org/plosbiology/s/figures#loc-blot-and-gel-reporting-requirements

>>Please provide the raw, unadjusted and uncropped images supporting Fig 1A-Q; Figure S1A;

2) BLURB: Please provide a blurb which (if accepted) will be included in our weekly and monthly Electronic Table of Contents, sent out to readers of PLOS Biology, and may be used to promote your article in social media. The blurb should be about 30-40 words long and is subject to editorial changes. It should, without exaggeration, entice people to read your manuscript. It should not be redundant with the title and should not contain acronyms or abbreviations

3) DATA POLICY:

a. Supplementary files (e.g., excel). Please ensure that all data files are uploaded as 'Supporting Information' and are invariably referred to (in the manuscript, figure legends, and the Description field when uploading your files) using the following format verbatim: S1 Data, S2 Data, etc. Multiple panels of a single or even several figures can be included as multiple sheets in one excel file that is saved using exactly the following convention: S1_Data.xlsx (using an underscore).

b. Deposition in a publicly available repository. Please also provide the accession code or a reviewer link so that we may view your data before publication. 

>>Regardless of the method selected, please ensure that you provide the individual numerical values that underlie the summary data displayed in the following figure panels as they are essential for readers to assess your analysis and to reproduce it:

Fig 1A,C-H,J-K; 2B,D,F-H,J-L,N-P; Fig 3; Fig 4; Fig 5; Fig 6B,D,J,F-H,L,N-P,R; Fig7D,F-L,N,P,R-S

Fig S1B-E; S2C,E,H,J; S3A-K,N-V; S4; S5C-E; S6B-K; S7B-H;

>>Please also ensure that figure legends in your manuscript include information on where the underlying data can be found, and ensure your supplemental data file/s has a legend.

>>Please ensure that your Data Statement in the submission system accurately describes where your data can be found.

We expect to receive your revised manuscript within two weeks. 

- a Response to Editors file that provides a detailed response to the the requests, above

*Published Peer Review History*

*Press*

Sincerely,

Lucas

Lucas Smith, Ph.D.

Associate Editor,

lsmith@plos.org,

PLOS Biology

---

## [Editor Report · Decision Letter 3]

7 Nov 2022

Dear Dr Sigrist,

Thank you for the submission of your revised Research Article "A brain-wide form of presynaptic active zone plasticity orchestrates resilience to brain aging in Drosophila" for publication in PLOS Biology. On behalf of my colleagues and the Academic Editor, Paul J Shaw, I am pleased to say that we can now, in principle, accept your manuscript for publication, provided you address any remaining formatting and reporting issues. These will be detailed in an email you should receive within 2-3 business days from our colleagues in the journal operations team; no action is required from you until then. Please note that we will not be able to formally accept your manuscript and schedule it for publication until you have completed any requested changes.

PRESS

Sincerely, 

Lucas Smith, Ph.D., Ph.D.

Associate Editor

PLOS Biology

lsmith@plos.org